# Improving Adversarial Robustness by Contrastive Guided Diffusion Process

## Abstract

Synthetic data generation has become an emerging tool to help improve the adversarial robustness in classification tasks since robust learning requires a significantly larger amount of training samples compared with standard classification tasks. Among various deep generative models, the diffusion model has been shown to produce high-quality synthetic images and has achieved good performance in improving the adversarial robustness. However, diffusion-type methods are typically slow in data generation as compared with other generative models. Although different acceleration techniques have been proposed recently, it is also of great importance to study how to improve the sample efficiency of generated data for the downstream task. In this paper, we first analyze the optimality condition of synthetic distribution for achieving non-trivial robust accuracy. We show that enhancing the distinguishability among the generated data is critical for improving adversarial robustness. Thus, we propose the Contrastive-Guided Diffusion Process (Contrastive-DP), which adopts the contrastive loss to guide the diffusion model in data generation. We verify our theoretical results using simulations and demonstrate the good performance of Contrastive-DP on image datasets.

## 1 Introduction

The success of most deep learning methods relies heavily on a massive amount of training data, which can be expensive to acquire in practice. For example, in autonomous driving (O'Kelly et al., 2018) and the medical diagnosis (Das et al., 2022) type applications, the number of rare scenes is usually very limited in real data. Moreover, it may be expensive to label the data in supervised learning. These challenges call for methods that can produce additional training data that satisfy two essential properties: (i) the additional data should help improve the downstream task performance; (ii) the additional data should be easy to generate. Synthetic data generation based on deep generative models has shown promising performance recently to tackle these challenges (Sehwag et al., 2022; Gowal et al., 2021; Das et al., 2022).

In synthetic data generation, one aims to learn a *synthetic distribution* (from which we generate synthetic data) that is close to the true date-generating distribution based on training data available, and most importantly, can help improve the downstream task performance. Synthetic data generation is highly related to generative models. Among various kinds of generative models, the score-based model and diffusion type models have gained much success in image generation recently (Song & Ermon, 2019; Song et al., 2021b; 2020; Song & Ermon, 2020; Sohl-Dickstein et al., 2015; Nichol & Dhariwal, 2021; Bao et al., 2022; Rombach et al., 2022). As validated in image datasets, the prototype of diffusion models, the Denoising Diffusion Probabilistic Model (DDPM) (Ho et al., 2020), and many variants can generate high-quality image data as compared with classical generative models such as GANs (Dhariwal & Nichol, 2021).

This paper mainly focuses on the adversarial robust classification of image data, which typically requires more training data than standard classification tasks. In Gowal et al. (2021), 100M high-quality synthetic images are generated by DDPM and achieve the state-of-the-art performance on adversarial robustness on the CIFAR-10 dataset, which demonstrates the effectiveness of diffusion models in improving adversarial robustness. However, a major drawback of diffusion-type methods is the slow computational speed. More specifically, DDPM is usually 1000 times slower than GAN (Song et al., 2021a) and this drawback is more serious when generating a large number of samples,

e.g., it takes more than 99 GPU days [1] for generating 100M image data according to Gowal et al. (2021). Moreover, the computational costs will also increase dramatically when the resolution of images increases, which inspires a plentiful of works studying how to accelerate the diffusion models (Song et al., 2021a; Watson et al., 2022; Ma et al., 2022; Salimans & Ho, 2022; Bao et al., 2022; Cao et al., 2022; Yang et al., 2022). In this paper, we aim to study the aforementioned problem from a different perspective – "how to generate effective synthetic data that are most helpful for the downstream task?". We analyze the *optimal synthetic distribution* for the downstream tasks to improve the sample efficiency of the generative model.

We first study the theoretical insights for finding the optimal synthetic distributions for achieving adversarial robustness. Following the setting considered in Carmon et al. (2019), we introduce a family of synthetic distributions controlled by the distinguishability of the representation from different classes. Our theoretical results show that the more distinguishable the representation is for the synthetic data, the higher the classification accuracy we will get when training a model on such synthetic data sets. Motivated by the theoretical insights, we propose the Contrastive-Guided Diffusion Process (Contrastive-DP) for efficient synthetic data generation, incorporating the contrastive learning loss (van den Oord et al., 2018; Chuang et al., 2020; Robinson et al., 2021) into the diffusion process. We conduct comprehensive simulations and experiments on real image datasets to demonstrate the effectiveness of the proposed Contrastive-DP.

The remainder of the paper is organized as follows. Section 2 presents the problem formulation and preliminaries on diffusion models. Section 3 contains the theoretical insights of optimal synthetic distribution under the Gaussian setting. Section 4 proposes a new type of data generation procedure that combines contrastive learning with diffusion models, as motivated by the theoretical insights obtained in Section 3. Finally, Section 5 conducts extensive numerical experiments to validate the good performance of the proposed generation method on simulation and image datasets.

## 2   Problem formulation and preliminaries

We first give a brief overview of adversarial robust classification, which is our main focus, but the whole framework is widely applicable to other downstream tasks in general. Denote the feature space as $\mathcal{X}$, the corresponding label space as $\mathcal{Y}$, and the true (joint) data distribution as $\mathcal{D} = \mathcal{D}_{\mathcal{X} \times \mathcal{Y}}$. Assume we have labeled training data $\mathcal{D}_{\text{train}} := \{(\boldsymbol{x}_i, y_i)\}_{i=1}^n$. We aim to learn a robust classifier $f_{\boldsymbol{\theta}} : \mathcal{X} \mapsto \mathcal{Y}$, parameterized by a learnable $\boldsymbol{\theta}$, that can achieve minimum adversarial loss:

$$\min_{\boldsymbol{\theta}} \mathcal{L}_{adv}(\boldsymbol{\theta}) := \mathbb{E}_{(\boldsymbol{x},y)\sim\mathcal{D}} \left( \max_{\boldsymbol{\delta}\in\Delta} \ell(\boldsymbol{x} + \boldsymbol{\delta}, y, \boldsymbol{\theta}) \right), \tag{1}$$

where $\ell(\boldsymbol{x}, y, \boldsymbol{\theta}) = 1\{y \neq f_\theta(\boldsymbol{x})\}$ is the 0-1 loss function, $1\{\cdot\}$ is the indicator function, and $\Delta = \{\boldsymbol{\delta} : \|\boldsymbol{\delta}\|_\infty \leq \epsilon\}$ is the adversarial set defined using $\ell_\infty$-norm. Intuitively, the solution to (1) is a robust classifier that minimizes the worst-case loss within an $\epsilon$-neighborhood of the input features.

In the canonical form of adversarial training, we train the robust classifier $f_{\boldsymbol{\theta}}$ on the training set $\mathcal{D}_{\text{train}} := \{(\boldsymbol{x}_i, y_i)\}_{i=1}^n$ by solving the following sample average approximation of (1):

$$\min_{\boldsymbol{\theta}} \widehat{\mathcal{L}}_{adv}(\boldsymbol{\theta}) := \frac{1}{n} \sum_{i=1}^n \max_{\boldsymbol{\delta}_i\in\Delta} \ell(\boldsymbol{x}_i + \boldsymbol{\delta}_i, y_i, \boldsymbol{\theta}). \tag{2}$$

### 2.1   Adversarial training using synthetic data

Synthetic data generation is one way to artificially increase the size of the training set by generating a sufficient amount of additional data, thus helping improve the learning algorithm's performance (Gowal et al., 2021). The mainstream generation procedures can be categorized into two types: (i) generate the features ($\boldsymbol{x}$) first and then assign pseudo labels to the generated features; (ii) or perform conditional generation conditioned on the desired label. Our analysis is mainly based on the former paradigm, which can be easily generalized to the conditional generation procedure, and our proposed algorithm is also flexible enough for both pipelines. Denote the distribution of the generated features as $\widetilde{\mathcal{D}}_{\mathcal{X}}$ and the generated synthetic data as $\mathcal{D}_{\text{syn}} := \{(\tilde{\boldsymbol{x}}_i, \tilde{y}_i)\}_{i=1}^{\tilde{n}}$. Here the feature

---

[1] Running on a RTX 4x2080Ti GPU cluster.

values $\tilde{\boldsymbol{x}}_i$ are generated from the synthetic distribution $\widetilde{\mathcal{D}}_{\mathcal{X}}$, and $\tilde{y}_i$ are pseudo labels assigned by a classifier learned on the training data $\mathcal{D}_{\text{train}}$. Combining the synthetic and real data, we will learn the robust classifier using a larger training set $\mathcal{D}_{\text{all}} := \mathcal{D}_{\text{train}} \cup \mathcal{D}_{\text{syn}}$ which now contains $n + \tilde{n}$ samples:

$$\min_{\boldsymbol{\theta}} \left\{ \eta \left( \frac{1}{n} \sum_{i=1}^{n} \max_{\boldsymbol{\delta}_i \in \Delta} \ell(\boldsymbol{x}_i + \boldsymbol{\delta}_i, y_i, \boldsymbol{\theta}) \right) + (1 - \eta) \left( \frac{1}{\tilde{n}} \sum_{i=1}^{n} \max_{\boldsymbol{\delta}_i \in \Delta} \ell(\tilde{\boldsymbol{x}}_i + \boldsymbol{\delta}_i, \tilde{y}_i, \boldsymbol{\theta}) \right) \right\}, \quad (3)$$

where $\eta \in (0, 1)$ is a parameter controlling the weights of synthetic data.

## 2.2 DIFFUSION MODEL FOR SYNTHETIC DATA GENERATION

We build our proposed generation procedure based on the Denoising Diffusion Probabilistic Model (DDPM) (Ho et al., 2020) and its accelerated variant Denoising Diffusion Implicit Model (DDIM) (Song et al., 2021a). In the following, we briefly review the key components of DDPM.

The core of DDPM is a forward Markov chain with Gaussian transitions $q(\boldsymbol{x}_t|\boldsymbol{x}_{t-1})$ to inject noise to the original data distribution $q(\boldsymbol{x}_0)$. More specifically, Ho et al. (2020) model the forward Gaussian transition as:

$$q\left(\boldsymbol{x}_t|\boldsymbol{x}_{t-1}\right) := \mathcal{N}\left(\sqrt{\alpha_t}\boldsymbol{x}_{t-1}, (1 - \alpha_t)\,\mathbb{I}\right),$$

where $\alpha_t, t = 1, 2, \ldots, T$ is a decreasing sequence to control the variance of injected noise, and $\mathbb{I}$ is the identity covariance matrix. The joint likelihood for the above Markov chain can be written as $q\left(\boldsymbol{x}_{0:T}\right) = q\left(\boldsymbol{x}_0\right) \prod_{t=1}^{T} q\left(\boldsymbol{x}_t|\boldsymbol{x}_{t-1}\right)$. DDPM then assumes we have $p_\theta\left(\boldsymbol{x}_{0:T}\right) = p_\theta\left(\boldsymbol{x}_T\right) \prod_{t=1}^{T} p_\theta\left(\boldsymbol{x}_{t-1}|\boldsymbol{x}_t\right)$ for the reverse process, where $p_\theta(\boldsymbol{x}_{t-1}|\boldsymbol{x}_t)$ is parameterized using a neural network. The training objective is to minimize the Kullback–Leibler (KL) divergence between the forward and reverse process, $\mathrm{D}_{\text{KL}}(q\left(\boldsymbol{x}_{0:T}\right), p_\theta\left(\boldsymbol{x}_{0:T}\right))$, which can be simplified as:

$$\min_{\theta} \mathbb{E}_{t,\boldsymbol{x}_0,\epsilon} \left[ \left\| \boldsymbol{\epsilon} - \boldsymbol{\epsilon}_\theta \left( \sqrt{\bar{\alpha}_t}\boldsymbol{x}_0 + \sqrt{1 - \bar{\alpha}_t}\boldsymbol{\epsilon}, t \right) \right\|^2 \right],$$

where $\boldsymbol{x}_0 \sim q(\boldsymbol{x}_0)$, $\bar{\alpha}_t = \prod_{s=1}^{t} \alpha_s$ for $t = 1, \ldots, T$, $\boldsymbol{\epsilon} \sim \mathcal{N}(\mathbf{0}, \mathbb{I})$, and $\boldsymbol{\epsilon}_\theta(\boldsymbol{x}, t)$ denotes the neural network parameterized by $\boldsymbol{\theta}$ to be learned. We refer to Ho et al. (2020) for the detailed algorithms.

After learning the time-reversed process parameterized by $\theta$, the original generation process in Ho et al. (2020) is a time-reversed Markov chain as follows:

$$\boldsymbol{x}_{t-1} = \frac{1}{\sqrt{\alpha_t}} \left( \boldsymbol{x}_t - \frac{1 - \alpha_t}{\sqrt{1 - \bar{\alpha}_t}} \boldsymbol{\epsilon}_\theta \left( \boldsymbol{x}_t, t \right) \right) + \sigma_t \boldsymbol{z}_t, \quad t = T, T - 1, \ldots, 1,$$

where $\boldsymbol{z}_t \sim \mathcal{N}(\mathbf{0}, \mathbb{I})$ if $t > 1$ and $\boldsymbol{z}_t = \mathbf{0}$ if $t = 1$. DDIM (Song et al., 2021a) speeds up the above procedure by generalizing the diffusion process to a non-Markovian process, leading to a sampling trajectory much shorter than $T$. DDIM carefully designs the forward transition $q(\boldsymbol{x}_{t-1}|\boldsymbol{x}_t, \boldsymbol{x}_0)$ such that $q\left(\boldsymbol{x}_t|\boldsymbol{x}_0\right) = \mathcal{N}\left(\sqrt{\bar{\alpha}_t}\boldsymbol{x}_0, (1 - \alpha_t)\mathbb{I}\right)$ for all $t = 1, \ldots, T$. The great advantage of DDIM is that it admits the same training objective as DDPM, which means we can adapt the pre-trained model of DDPM and accelerate the sampling process without additional cost. The key sample-generating step in DDIM is as follows:

$$\boldsymbol{x}_{t-1} = \sqrt{\alpha_{t-1}} \underbrace{\left( \frac{\boldsymbol{x}_t - \sqrt{1 - \alpha_t}\boldsymbol{\epsilon}_\theta\left(\boldsymbol{x}_t, t\right)}{\sqrt{\alpha_t}} \right)}_{\text{predicted } \boldsymbol{x}_0} + \underbrace{\sqrt{1 - \alpha_{t-1}} \cdot \boldsymbol{\epsilon}_\theta\left(\boldsymbol{x}_t, t\right)}_{\text{pointing to } \boldsymbol{x}_t}, \quad (4)$$

in which we can generate $\boldsymbol{x}_{t-1}$ using $\boldsymbol{x}_t$ and $\boldsymbol{x}_0$. Also, the generating process becomes deterministic.

## 3 THEORETICAL INSIGHTS: OPTIMAL SYNTHETIC DISTRIBUTION

In this section, we consider a concrete distributional model as used in Carmon et al. (2019); Schmidt et al. (2018), and demonstrate the advantage of refining the synthetic data generation process – using the optimal distribution for synthetic data generation can help reduce the sample complexity needed for robust classification. This provides theoretical insights and motivates the proposed generation method to be introduced in Section 4.

### 3.1 THEORETICAL SETUP

Consider a binary classification task where $\mathcal{X} = \mathbb{R}^d, \mathcal{Y} = \{-1, 1\}$. The true data distribution $\mathcal{D} = \mathcal{D}_{\mathcal{X} \times \mathcal{Y}}$ is specified as follows. The marginal distribution for label $y$ is uniform in $\mathcal{Y}$, and the conditional distribution of features is $\boldsymbol{x}|y \sim \mathcal{N}(y\boldsymbol{\mu}, \sigma^2 \mathbb{I}_d)$, where $\boldsymbol{\mu} \in \mathbb{R}^d$ is non-zero, and $\mathbb{I}_d$ is the $d$ dimensional identity covariance matrix. Assume we generate a set of synthetic data from another synthetic distribution $\widetilde{\mathcal{D}}$.

We focus on learning a robust linear classifier under such setting. The family of linear classifiers is represented as $f_{\boldsymbol{\theta}}(\boldsymbol{x}) = \text{sign}(\boldsymbol{\theta}^\top \boldsymbol{x})$. Recall that we first generate features and then assign pseudo labels to the features. Therefore, a self-learning paradigm is adopted here (Wei et al., 2020). Given a set of unlabeled features $\{\tilde{\boldsymbol{x}}_1, \tilde{\boldsymbol{x}}_2, \ldots, \tilde{\boldsymbol{x}}_{\tilde{n}}\}$, we apply an intermediate linear classifier parameterized by $\hat{\boldsymbol{\theta}}_{\text{inter}} = \frac{1}{n}\sum_{i=1}^n y_i \boldsymbol{x}_i$, learned from real data $\mathcal{D}_{\text{train}}$, to assign the pseudo-label. Then, the synthetic data $\mathcal{D}_{\text{syn}} = \{(\tilde{\boldsymbol{x}}_1, \tilde{y}_1), \ldots, (\tilde{\boldsymbol{x}}_{\tilde{n}}, \tilde{y}_{\tilde{n}})\}$, where $\tilde{y}_i = \text{sign}(\hat{\boldsymbol{\theta}}_{\text{inter}}^\top \boldsymbol{x}_i), i = 1, \ldots, \tilde{n}$. We combine the real data and synthetic data $\mathcal{D}_{\text{all}} := \mathcal{D}_{\text{train}} \cup \mathcal{D}_{\text{syn}} = \{\{(\boldsymbol{x}_i, y_i)\}_{i=1}^n, \{(\tilde{\boldsymbol{x}}_i, \tilde{y}_i)\}_{i=1}^{\tilde{n}}\}$ to obtain an approximate optimal solution $\hat{\boldsymbol{\theta}}_{\text{final}}$ as:

$$\hat{\boldsymbol{\theta}}_{\text{final}} = \frac{1}{n + \tilde{n}}\Big(\sum_{i=1}^n y_i \boldsymbol{x}_i + \sum_{j=1}^{\tilde{n}} \tilde{y}_j \tilde{\boldsymbol{x}}_j\Big). \tag{5}$$

Note that the final linear classifier $\hat{\boldsymbol{\theta}}_{\text{final}}$ depends on the synthetic data generated from $\widetilde{\mathcal{D}}$. We aim to study which synthetic distribution $\widetilde{\mathcal{D}}$ can help reduce the adversarial classification error (also called robust error) $\text{err}_{\text{robust}}(f_{\hat{\boldsymbol{\theta}}_{\text{final}}}) := \mathbb{P}_{(\boldsymbol{x},y)\sim\mathcal{D}}(\exists \boldsymbol{\delta} \in \Delta, f_{\hat{\boldsymbol{\theta}}_{\text{final}}}(\boldsymbol{x} + \boldsymbol{\delta}) \neq y)$, where $\Delta = \{\boldsymbol{\delta} : \|\boldsymbol{\delta}\|_\infty \leq \epsilon\}$. And we similarly define the standard error as $\text{err}_{\text{standard}}(f_{\hat{\boldsymbol{\theta}}_{\text{final}}}) := \mathbb{P}_{(\boldsymbol{x},y)\sim\mathcal{D}}(f_{\hat{\boldsymbol{\theta}}_{\text{final}}}(\boldsymbol{x}) \neq y)$ which will be used later.

**Remark 1** (Comparison with existing literature). *In Carmon et al. (2019); Deng et al. (2021), sample complexity results are proposed based on the same Gaussian mixture setting. The major difference is that they all assume the learned linear classifier $\hat{\boldsymbol{\theta}}_{final}$ is only learned from synthetic data $\mathcal{D}_{syn}$ rather than the combination of the real and synthetic data $\mathcal{D}_{all}$. In general, our theoretical setup matches well with the practical algorithms.*

### 3.2 THEORETICAL INSIGHTS FOR OPTIMAL SYNTHETIC DISTRIBUTION

We first study the desired properties of the synthetic distribution $\widetilde{\mathcal{D}}$ that can lead to a better adversarial classification accuracy when the additional synthetic sample $\mathcal{D}_{\text{syn}}$ is used in the training stage. In Carmon et al. (2019), the case $\widetilde{\mathcal{D}} = \mathcal{D}$ is studied, i.e., they consider the case that additional unlabeled data from the true distribution $\mathcal{D}$ is available, and they characterize the usefulness of those additional training data. Compared with from Carmon et al. (2019), we consider general distributions $\widetilde{\mathcal{D}}$ which does not necessarily equal to $\mathcal{D}$.

First note that by the Bayes rule, the optimal decision boundary for the true data distribution is given by $\boldsymbol{\mu}^\top \boldsymbol{x} = 0$. Therefore, we restrict our attention to synthetic data distributions that satisfy: (i) the marginal distribution of the label $\tilde{y}$ is also uniform in $\mathcal{Y}$, same as $\mathcal{D}$; (ii) the conditional probability densities $p(\tilde{\boldsymbol{x}}|\tilde{y} = 1)$ and $p(\tilde{\boldsymbol{x}}|\tilde{y} = -1)$ of the synthetic data distribution are symmetric around the true optimal decision boundary $\boldsymbol{\mu}^\top \boldsymbol{x} = 0$. More specifically, we start with a special case of the synthetic data distribution $\widetilde{\mathcal{D}}_{\mathcal{X}} = 0.5\mathcal{N}(\tilde{\boldsymbol{\mu}}, \sigma^2 \mathbb{I}) + 0.5\mathcal{N}(-\tilde{\boldsymbol{\mu}}, \sigma^2 \mathbb{I})$ (note that when $\tilde{\boldsymbol{\mu}} = c\boldsymbol{\mu}$ for some constant $c$, the above two conditions are all satisfied).

In the following proposition, we present several representative scenarios of synthetic distributions in terms of how they may contribute to the downstream classification task. Figure 1 gives a pictorial demonstration for different cases.

**Proposition 1.** *Consider a special form of synthetic distributions $\widetilde{\mathcal{D}}_{\mathcal{X}} = 0.5\mathcal{N}(\tilde{\boldsymbol{\mu}}, \sigma^2 \mathbb{I}) + 0.5\mathcal{N}(-\tilde{\boldsymbol{\mu}}, \sigma^2 \mathbb{I})$ and assume $\{\tilde{\boldsymbol{x}}_1, \ldots, \tilde{\boldsymbol{x}}_{\tilde{n}}\}$ are samples from $\widetilde{\mathcal{D}}_{\mathcal{X}}$. We follow the self-learning paradigm described in Section 3.1 to learn the classifier $f_{\hat{\boldsymbol{\theta}}_{final}}$, when $\tilde{n}$ is sufficiently large we have:*

*Case 1: Inefficient $\widetilde{\mathcal{D}}_{\mathcal{X}}$. When $\langle \tilde{\boldsymbol{\mu}}, \boldsymbol{\mu} \rangle = 0$, the standard error $\text{err}_{standard}(f_{\hat{\boldsymbol{\theta}}_{final}})$ achieves the maximum and when $\langle \tilde{\boldsymbol{\mu}}, \boldsymbol{\mu} - \varepsilon \mathbf{1}_d \rangle = 0$, the robust error $\text{err}_{robust}(f_{\hat{\boldsymbol{\theta}}_{final}})$ achieves the maximum.*

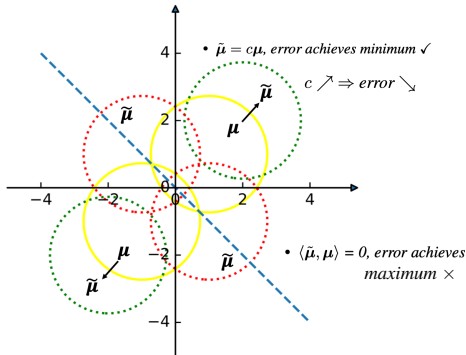

Figure 1: Demonstration of Proposition 1.

*Case 2: Optimal $\widetilde{\mathcal{D}}_{\mathcal{X}}$ for clean accuracy. When $\tilde{\boldsymbol{\mu}} = c\boldsymbol{\mu}$ for $c > 0$, $\mathrm{err}_{standard}\left(f_{\hat{\boldsymbol{\theta}}_{final}}\right)$ achieves the minimum, and the bigger the $c$ is, the smaller the $\mathrm{err}_{standard}\left(f_{\hat{\boldsymbol{\theta}}_{final}}\right)$.*

*Case 3: Optimal $\widetilde{\mathcal{D}}_{\mathcal{X}}$ for robust accuracy. When $\tilde{\boldsymbol{\mu}} = c(\boldsymbol{\mu} - \varepsilon\mathbf{1}_d)$ for $c > 0$, the robust error $\mathrm{err}_{robust}\left(f_{\hat{\boldsymbol{\theta}}_{final}}\right)$ achieves the minimum, and the bigger the $c$ is, the smaller the $\mathrm{err}_{robust}\left(f_{\hat{\boldsymbol{\theta}}_{final}}\right)$.*

**Remark 2** (Comparison with the existing characterization of the synthetic distribution). *We briefly comment on the main differences and similarities with Deng et al. (2021), in which a similar result was presented in Theorem 4. In Deng et al. (2021), the optimal solution of $\boldsymbol{\theta}^*$ was given for minimizing robust error $\mathrm{err}_{robust}\left(f_{\hat{\boldsymbol{\theta}}_{final}}\right)$ and they provides a specific unlabeled distribution $\tilde{\boldsymbol{\mu}} = \boldsymbol{\mu} - \varepsilon\mathbf{1}_d$ that achieves asymptotic optimality under certain condition. In this paper, we propose a general family of optimal distribution controlled by a scalar $c$, which represents the distinguishability of the feature. The optimal $\boldsymbol{\theta}^*$ proposed in Deng et al. (2021) can be recovered when $c \to \infty$. Therefore, our conclusion points out the optimality condition of unlabeled distribution and inspires a line of work to improve the performance of $\hat{\boldsymbol{\theta}}_{final}$ by making the feature of unlabeled distribution distinguishable.*

Table 1: Simulation results validating findings in Proposition 1. $d = 2$ and $d = 100$ denotes the dimension of $\boldsymbol{x}$, representing the low-dimensional and high-dimensional cases, respectively. For $d = 2$, we set $\|\boldsymbol{\mu}\|^2 = 2$, $\varepsilon = 0.5$, and for $d = 100$, we set $\|\boldsymbol{\mu}\|^2 = 4$, $\varepsilon = 0.1$. We use "Real" to denote the real data distribution and $n$ to denote the number of data from the real distribution, while we use "$c$" to denote different synthetic distributions with $\tilde{\boldsymbol{\mu}} = c\boldsymbol{\mu}$ and use $\tilde{n}$ to denote the number of synthetic data. The results and the standard deviation in the bracket are averaged over 50 independent trials.

| | | d=2 | | | | d=100 | |
| --- | --- | --- | --- | --- | --- | --- | --- |
| | | Accuracy | Robust Accuracy | | | Accuracy | Robust Accuracy |
| Real | $n = 10$ | 0.9201 (0.0012 ) | **0.7593 (0.0020)** | $n = 10$ | | 0.9023 (0.0192) | 0.6843 (0.0359) |
| | $n = 100$ | 0.9205 (0.0004) | **0.7608 (0.0006)** | $n = 100$ | | 0.9682 (0.0014) | 0.8239 (0.0028) |
| $c = 0.5$ | $\tilde{n} = 10$ | 0.9159 (0.0099 ) | 0.7541 (0.0096) | $\tilde{n} = 10$ | | 0.7562 (0.0564) | 0.4611 (0.0694) |
| | $\tilde{n} = 100$ | 0.9213 (0.0011) | 0.7601 (0.0009) | $\tilde{n} = 100$ | | 0.9505 (0.0047) | 0.7848 (0.0111) |
| $c = 1$ | $\tilde{n} = 10$ | 0.9133 (0.0066) | 0.7502 (0.0061) | $\tilde{n} = 10$ | | 0.8866 (0.0273) | 0.6557 (0.0487) |
| | $\tilde{n} = 100$ | 0.9165 (0.0005) | 0.7528 (0.0006) | $\tilde{n} = 100$ | | 0.9695 (0.0012) | 0.8239 (0.0031) |
| $c = 1.5$ | $\tilde{n} = 10$ | **0.9209** (0.0038) | 0.7523 (0.0025) | $\tilde{n} = 10$ | | **0.9400** (0.0100) | **0.7603** (0.0233) |
| | $\tilde{n} = 100$ | **0.9232** (0.0003) | 0.7538 (0.0005) | $\tilde{n} = 100$ | | **0.9743** (0.0008) | **0.8343** (0.0011) |

**Simulation results.** To verify the findings in Proposition 1, we conduct extensive simulation experiments for a Gaussian example with varying data dimensions, sample sizes, and the position of $\tilde{\boldsymbol{\mu}}$. In Table 1, we demonstrate the clean and robust accuracy learned on synthetic distribution with the angle between $\boldsymbol{\mu}$ and $\epsilon\mathbf{1}_d$ equals $0°$. More experiment results (with $30°$, $60°$, and $90°$) can be found in Table 7, 8, 9 and 10 in Appendix B. In most cases, the classifier learned from the synthetic distribution with $\tilde{\boldsymbol{\mu}} = c\boldsymbol{\mu}$ with $c > 1$ achieves better performance even than the iid samples.

# 4 DIFFUSION MODELS GUIDED BY CONTRASTIVE LOSS

Proposition 1 and the corresponding simulation results in Table 1 show that the synthetic data can help improve the classification task especially when the representation of different classes is more distinguishable in the synthetic distribution. Therefore, contrastive loss (van den Oord et al., 2018) can be adopted to explicitly control the distances of the representation of different classes. Therefore, we propose a variant of the classical diffusion model, named *Contrastive-Guided Diffusion Process (Contrastive-DP)*, to enhance the sample efficiency of the generative model. In this section, we first present the overall algorithm of the proposed Contrastive-DP procedure in Section 4.1, then we describe the detailed design of the contrastive loss in Section 4.2.

## 4.1 CONTRASTIVE-GUIDED DIFFUSION PROCESS

The detailed generation procedure of Contrastive-DP is given in Algorithm 1. We highlight below some major differences between the proposed Contrastive-DP and the vanilla DDIM algorithm. In each time step $t$ of the generation procedure, given the current value $x_t^{(i)}$, we add the gradient of the contrastive loss $\ell_{\text{contra}}(x_t^{(i)}, x_p^{(i)}; \tau)$ with respect to $x_t^{(i)}$ to the original diffusion generative process, here $x_p^{(i)}$ is the positive pair of $x_t^{(i)}$ (will be explained in detail later), $\tau$ is the temperature for softmax, and $\lambda$ is the hyperparameter balancing the contrastive loss within the diffusion process.

This modification ensures that the generated data will be distinguishable among data in the same batch. The construction of the contrastive loss $\ell_{\text{contra}}(\cdot)$ is very flexible – we can adopt multiple forms of contrastive loss together with different selection strategies of positive and negative pairs, which will be discussed in detail in the following.

## 4.2 CONTRASTIVE LOSS FOR DIFFUSION PROCESS

---

**Algorithm 1** Generation in Contrastive-guided Diffusion Process (Contrastive-DP)

---

1: $\mathbb{X}_T = \{x_T^{(i)}\}_{i=1}^m \sim \mathcal{N}(\mathbf{0}, \mathbb{I})$
2: $t = T$
3: **while** $t \neq 1$ **do**
4:     **for** i = 1:m **do**
5:         Choosing $x_p^{(i)}$ as the positive pair of $x_t^{(i)}$
6:         $\Delta x_t^{(i)} = \lambda \cdot \nabla_{x_t^{(i)}} \ell_{\text{contra}}(x_t^{(i)}, x_p^{(i)}; \tau) + \epsilon_\theta(x_t^{(i)}, t)$
7:         $x_{t-1}^{(i)} = \sqrt{\alpha_{t-1}}\left(\frac{x_t^{(i)} - \sqrt{1-\alpha_t} \Delta x_t^{(i)}}{\sqrt{\alpha_t}}\right) + \sqrt{1 - \alpha_{t-1}} \cdot \Delta x_t^{(i)}$
8:         $t = t - 1$
9:     **end for**
10: **end while**
11: **return** $\mathbb{X}_0 = \{x_0^{(i)}\}_{i=1}^m$

---

Let $\mathbb{X} = x_1, ..., x_m$ be a minibatch of training data. We apply the contrastive loss to the embedding space. Assume $f(\cdot)$ is the feature extractor that maps the input data in $\mathbb{X}$ onto the embedding space. In general, we adopt two forms of the contrastive loss $\ell_{\text{contra}}(x_t^{(i)}, x_p^{(i)}; \tau)$ used in Algorithm 1.

First is the InfoNCE loss: $\ell_{\text{InfoNCE}}(x_a, x_p; \tau) = -\log(g_\tau(x_a, x_p)/\sum_{k=1}^m \mathbf{1}_{k \neq a} g_\tau(x_a, x_k))$, where $m$ is the batch size, $\tau$ is the temperature for softmax, $x_a$, $x_p$ denote the anchor and the positive pair, respectively, $g_\tau(x, x') = \exp(f(x)^\top f(x')/\tau)$, and all images except the anchor $x_a$ in the minibatch $\mathbb{X}$ is negative pairs. InfoNCE loss is an unsupervised learning metric and does not explicitly distinguish the representation from different classes, which implicitly regards the representation from the same class as negative pair.

Second is the hard negative mining loss: $\ell_{\text{HNM}}(x_a, x_p; \tau) = -\log(g_\tau(x_a, x_p)/(g_\tau(x_a, x_p) + m/\tau^- (\mathbb{E}_{x_n \sim q_\beta}[(g_\tau(x_a, x_n)] - \tau^+ \mathbb{E}_{v \sim q_\beta^+}[(g_\tau(x_a, v)])])))$, where $m$ denotes the batch size, $\tau^- = 1 - \tau^+$ denotes the probability of observing any different class with $x_a$ and $q_\beta$ is an unnormalized von Mises–Fisher distribution (Jammalamadaka, 2011), with mean direction $f(x)$ and "concentration parameter" $\beta$ to control the hardness of negative mining; $q_\beta$ and $q_\beta^+$ can be easily approximated

by Monte-Carlo importance sampling techniques. We refer to Chuang et al. (2020); Robinson et al. (2021) for detailed descriptions of hard negative mining contrastive loss. Compared with the InfoNCE loss that does not consider class/label information, the hard negative mining (HNM) loss enhances the discriminative ability of different classes in the feature space.

It is worth mentioning that the Contrastive-DP enjoys the plugin-type property – it does not modify the original training procedure of diffusion processes and can be easily adopted to various kinds of diffusion models.

**Numerical Validations.** We first demonstrate the effectiveness of contrastive-DP in Figure 2 using a simulation example. Consider the binary classification problem as in Section 3.1, and the real data for each class are generated from a Gaussian distribution. Figure 2(a) demonstrates the synthetic data generated by the vanilla diffusion model, which recovers the ground-truth Gaussian distribution well. When using the contrastive-DP procedure with HNM loss, we obtain the generated synthetic data as shown in Figure 2(b), which is more distinguishable with a much smaller variance.

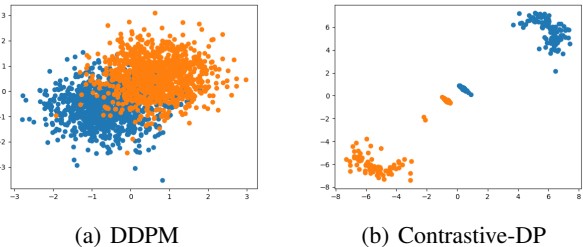

(a) DDPM        (b) Contrastive-DP

Figure 2: An illustration of the effectiveness of synthetic distribution guided by contrastive loss.

In addition, Figure 3 and Figure 4 in Appendix C.2 demonstrate the synthetic data distribution guided by different kinds of contrastive loss mentioned above. It can be shown that InfoNCE loss and hard negative mining method cannot explicitly distinguish the data within the same class and thus form a circle within each class to maximize the distance between samples, while the conditional version of contrastive loss (given the oracle class information) can make two classes more separable.

## 5 REAL-WORLD IMAGE DATASETS

In this section, we demonstrate the effectiveness of the proposed contrastive guided diffusion process for synthetic data generation in adversarial classification tasks. We first compare the performance of Contrastive-DP with the vanilla DDIM methods in Section 5.1. Then, we present a comprehensive ablation study on the performance of Contrastive-DP to shed insights on how to adopt the contrastive loss functions on the diffusion model in Section 5.2, especially on which contrastive loss gives the best performance on the diffusion process and how to choose the hyperparameter $\lambda$ to control the strength of the guidance of the contrastive loss.

### 5.1 EXPERIMENTAL RESULTS

We test the contrastive-DP algorithm on two image datasets, the CIFAR-10 dataset (Krizhevsky, 2009) and Traffic Signs dataset (Houben et al., 2013). CIFAR-10 dataset contains 50k training images in 10 classes and 10K images for testing, while Traffic signs dataset contains 39252 training images in 43 classes and 12629 images for testing. For CIFAR-10 dataset, we generate 50K, 200K, and 1M additional images together with the original training images for adversarial training, while for Traffic signs dataset, we synthetic 50k images. To demonstrate our Contrastive-DP algorithm is flexible to be adopted to various kinds of diffusion models and make use of the existing pretrained models, we establish our Contrastive-DP algorithm on the unconditional DDIM for CIFAR-10 datasets and on the conditional DDPM for Traffic signs dataset. The detailed description of the pipeline for generating data and the corresponding hyperparameter can be found in Appendix D.2.

Table 2 demonstrates the effectiveness of our contrastive-DP algorithm on the CIFAR-10 dataset [2], which achieves better robust accuracy on all data regimes than the vanilla DDIM. All of the results

---

[2]Since the Pytorch Implementation of Gowal et al. (2021) is not open source, we utilize the best unofficial implementation to reconduct all the experiments for a fair comparison.

are higher than the baseline result without synthetic data by a large margin (+4.37% in 50K setting and +7.3% in 1M setting). Table 3 demonstrates the effectiveness of our contrastive-DP algorithm on the Traffic Signs dataset. Our contrastive-DP achieves better clean and robust accuracy than the vanilla DDPM model and is also higher than the baseline result without synthetic data by a large margin (+9.96%).

Table 2: The clean and adversarial accuracy on the CIFAR-10 dataset. The robust accuracy is reported by the worst accuracy obtained by either AUTOATTACK (Croce & Hein, 2020) or AA+MT (Gowal et al., 2020) with $\epsilon_\infty = 8/255$ and WRN-28-10. 50k, 200k, and 1M denote the number of synthetic used for adversarial training.

| | No additional data | | 50K | | 200K | | 1M | |
|---|---|---|---|---|---|---|---|---|
| | clean acc | rob acc | clean acc | rob acc | clean acc | rob acc | clean acc | rob acc |
| WRN-28-10 (DDIM) | | | **84.65%** | 52.46% | **85.86%** | 54.99% | 85.37% | 56.61% |
| WRN-28-10 (Contrastive-DP) | 81.09% | 49.54% | 83.66% | **53.91%** | 85.71% | **55.79%** | **86.30%** | **56.84%** |

Table 3: The clean and adversarial accuracy on the Traffic Signs dataset. he results and the standard deviation in the bracket are averaged over 3 independent trials.

| | clean acc | rob acc |
|---|---|---|
| No additional data | 78.52% (0.16%) | 46.03%(0.85%) |
| DDPM | 86.79%(0.12%) | 56.01%(0.14%) |
| Contrastive-DP | **86.5%(0.62%)** | **56.27% (0.25%)** |

## 5.2 ABLATION STUDIES

**Sensitivity of** $\lambda$**.** Table 4 shows the influence of the strength of the contrastive loss. $\lambda = 100k$ gives consistently better results than a smaller $\lambda = 50k$ or a larger $\lambda = 200k$ on robust accuracy on all settings. Moreover, we find the larger the $\lambda$ is, the better performance we get on clean accuracy when the additional data is small (50K case), while the smaller the $\lambda$ is, the better performance we get on clean accuracy when the additional data is large (1M case).

Table 4: The performance of Contrastive-DP under different $\lambda$ values.

| | 50K | | 200K | | 1M | |
|---|---|---|---|---|---|---|
| | clean acc | rob acc | clean acc | rob acc | clean acc | rob acc |
| $\lambda = 50k$ | 84.41% | 53.78% | 85.45% | 55.24% | **86.35%** | 56.83% |
| $\lambda = 100k$ | 83.66% | **53.91%** | **85.71%** | **55.79%** | 86.30% | **56.84%** |
| $\lambda = 200k$ | **84.51%** | 53.55% | **85.51%** | 55.33% | 85.98% | 56.69% |

**The effectiveness of different contrastive losses.** Table 5 demonstrates the performance of different design of the contrastive loss. We find out that applying the hard negative mining together with the embedding network achieves better clean and robust accuracy when the additional data is small (50K and 200K setting), while the infoNCE loss achieves better clean and robust accuracy when the additional data is large (1M setting). This result shows that we can improve the sample efficiency of the generative model by carefully designing the contrastive loss.

Table 5: The performance of Contrastive-DP under different contrastive loss: infoNCE and HNM losses, and w/wo embedding denote with/without an embedding network.

| | 50K | | 200K | | 1M | |
|---|---|---|---|---|---|---|
| | clean acc | rob acc | clean acc | rob acc | clean acc | rob acc |
| DDIM+infoNCE | 83.40% | 52.74% | 84.18% | 54.75% | **85.64%** | **56.28%** |
| DDIM+HNM(w embedding) | **84.20%** | **53.19%** | **85.71%** | **54.92%** | 85.29% | 56.12% |
| DDIM+HNM(wo embedding) | 83.97% | 52.89% | 85.65% | 54.83% | 85.38% | 55.95% |

**Data selection for synthetic data.** Data selection methods are worthy of study since, in practice, we would like to know whether we can achieve better performance by generating a large number of samples and applying some selection criteria to filter out some samples. Therefore, we propose several data selection criterion and evaluate corresponding effectiveness in Table 6. All of the selection methods on Contrastive-DP are higher than vanilla DDIM plus selection methods, which

demonstrates the superiority of using the contrastive learning loss as the guidance rather than using selection methods on the images generated by the vanilla diffusion model.

Table 6: Comparison of different data selection criteria. The detailed explanation of each selection method can be found in Append D.3.

| | 50K | | 200K | | 1M | |
|---|---|---|---|---|---|---|
| | clean acc | rob acc | clean acc | rob acc | clean acc | rob acc |
| DDIM (Separability) | 79.93% | 49.49% | 85.09% | 54.90% | 84.87% | 56.08% |
| Contrastive-DP (Gradient norm) | 83.92% | 55.09% | 84.64% | 55.17% | **86.36%** | 57.11% |
| Contrastive-DP (Gradient norm-rob) | 83.91% | **55.23%** | 84.78% | 55.42% | 85.93% | **57.18%** |
| Contrastive-DP (Entropy) | **84.17%** | 55.08% | **85.71%** | **55.79%** | 86.30% | 56.84% |

## 6 RELATED WORK

Using generative models to improve adversarial robustness has attracted increasing attention recently. Gowal et al. (2021) uses 100M high-quality images generated by DDPM together with the original training set to achieve state-of-the-art performance on the CIFAR-10 dataset. They propose to use Complementary as an important metric for measuring the efficacy of the synthetic data. In Sehwag et al. (2022), it was claimed that the transferability of adversarial robustness between two data distributions is measured by conditional Wasserstein distance, which inspires us to use it as a criterion for selecting samples. Our work follows the same line, but we investigate how to generate the samples with high information rather than applying the selection to the data generated by the vanilla diffusion model. Below we also summarize some closely related work in different lines.

**Sample-efficient generation.** We can view the sample-efficient generation problem as a Bi-level optimization problem. We can regard how to synthesize data as the meta objective and the performance of the model trained on the synthetic data as the inner objective. For data-augmentation based methods, Ruiz et al. (2019) adopt a reinforcement learning based method for optimizing the generator in order to maximize the training accuracy. For active learning based methods, Tran et al. (2019) use an Auto-Encoder to generate new samples based on the informative training data selected by the acquisition function. Besides, Kim et al. (2020) combines the active learning criterion with data augmentation methods. They use the gradient of acquisition function after one-step augmentation as guidance for training the augmentation policy network.

**Theoretical analysis of adversarial robustness.** In Schmidt et al. (2018), the sample complexity of adversarial robustness has been shown to be substantially larger than standard classification tasks in the Gaussian setting. Carmon et al. (2019) bridges this gap by using the self-training paradigm and corresponding unlabeled data. Deng et al. (2021) further extends the aforementioned conclusion by leveraging out-of-domain unlabeled data. None of the works mentioned above investigates the optimal distribution for unlabeled synthetic data.

**Contrastive learning.** Contrastive learning algorithms have been widely used for representation learning (Chen et al., 2020; He et al., 2020; Grill et al., 2020). The vanilla contrastive learning loss, InfoNCE (van den Oord et al., 2018), aims to draw the distance between positive pairs and push the negative pairs away. To mitigate the problem that not all negative pairs may be true negatives, the negative hard mining criterion was proposed in (Chuang et al., 2020; Robinson et al., 2021).

## 7 CONCLUSION

In this paper, we delve into which kind of synthetic distribution is optimal for the downstream task, especially for achieving adversarial robustness in image data classification. We derive the optimality condition under the Gaussian setting and propose the Contrastive-guided Diffusion Process (Contrastive-DP), a plug-in algorithm suitable for various types of diffusion models. We verify our theorem on the Gaussian simulation and demonstrate the superiority of the Contrastive-DP algorithm on the image datasets.

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

## A   THEORETICAL DETAILS

### A.1   ERROR PROBABILITIES IN CLOSED FORM.

Here, we briefly recapitulate the closed form formulation for the standard and robust error probabilities as detailed in Carmon et al. (2019); Deng et al. (2021).

The standard error probability can be written as

$$\text{err}_{\text{standard}}\left(f_{\boldsymbol{\theta}}\right) = \mathbb{P}\left(y \cdot \boldsymbol{x}^{\top}\boldsymbol{\theta} < 0\right) = \mathbb{P}\left(\mathcal{N}\left(\frac{\boldsymbol{\mu}^{\top}\boldsymbol{\theta}}{\sigma\|\boldsymbol{\theta}\|}, 1\right) < 0\right) = Q\left(\frac{\boldsymbol{\mu}^{\top}\boldsymbol{\theta}}{\sigma\|\boldsymbol{\theta}\|}\right), \quad (6)$$

where

$$Q(x) = \frac{1}{\sqrt{2\pi}}\int_{x}^{\infty} e^{-t^2/2}dt$$

is the Gaussian error function and is non-increasing. Clearly the standard error probability is minimized when $\frac{\boldsymbol{\theta}}{\|\boldsymbol{\theta}\|} = \frac{\boldsymbol{\mu}}{\|\boldsymbol{\mu}\|}$, i.e., $\boldsymbol{\theta} = c\boldsymbol{\mu}$ for some scalar $c > 0$. We may impost $\|\boldsymbol{\theta}\|_2 = 1$ to ensure the unique solution $\boldsymbol{\theta} = \boldsymbol{\mu}/\|\boldsymbol{\mu}\|$.

The robust error probability under the $\ell_{\infty}$ adversarial set $\Delta = \{\delta : \|\boldsymbol{\delta}\|_{\infty} \le \epsilon\}$ is

$$\begin{aligned}
\text{err}_{\text{robust}}^{\infty,\varepsilon}\left(f_{\boldsymbol{\theta}}\right) &= \mathbb{P}\left(\inf_{\|\boldsymbol{\nu}\|_{\infty}\le\varepsilon}\left\{y\cdot(\boldsymbol{x}+\boldsymbol{\nu})^{\top}\boldsymbol{\theta}\right\} < 0\right) \\
&= \mathbb{P}\left(y\cdot\boldsymbol{x}^{\top}\boldsymbol{\theta} - \varepsilon\|\boldsymbol{\theta}\|_1 < 0\right) = \mathbb{P}\left(\mathcal{N}\left(\boldsymbol{\mu}^{\top}\boldsymbol{\theta}, \sigma^2\|\boldsymbol{\theta}\|^2\right) < \varepsilon\|\boldsymbol{\theta}\|_1\right) \\
&= Q\left(\frac{\boldsymbol{\mu}^{\top}\boldsymbol{\theta}}{\sigma\|\boldsymbol{\theta}\|} - \frac{\varepsilon\|\boldsymbol{\theta}\|_1}{\sigma\|\boldsymbol{\theta}\|}\right).
\end{aligned} \quad (7)$$

In the following part, we use a simpler notation $\text{err}_{\text{robust}}\left(f_{\boldsymbol{\theta}}\right)$ for the robust error $\text{err}_{\text{robust}}^{\infty,\varepsilon}\left(f_{\boldsymbol{\theta}}\right)$ without ambiguity. The closed-form of the optimal $\boldsymbol{\theta}^*$ that minimizes the above robust error $\text{err}_{\text{robust}}$ can be shown to be (Deng et al., 2021):

$$\boldsymbol{\theta}^* = \frac{T_{\varepsilon}(\boldsymbol{\mu})}{\|T_{\varepsilon}(\boldsymbol{\mu})\|},$$

where $T_{\varepsilon}(\boldsymbol{\mu})$ is the hard-thresholding operator with $(T_{\varepsilon}(\boldsymbol{\mu}))_j = \text{sign}\left(\boldsymbol{\mu}_j\right)\cdot\max\left\{|\boldsymbol{\mu}_j| - \varepsilon, 0\right\}$. Under the mild assumption $\boldsymbol{\mu}_j > \varepsilon, \forall j \in \{1, 2, \ldots, d\}$, the optimal solution can be simplified as:

$$\boldsymbol{\theta}^* = \frac{\boldsymbol{\mu} - \varepsilon\mathbf{1}}{\|\boldsymbol{\mu} - \varepsilon\mathbf{1}\|}.$$

**Remark 3.** *Note that when $\boldsymbol{\mu} = c\mathbf{1}$ for some constant $c > \epsilon$, the optimal solution $\boldsymbol{\theta}^* = \frac{\boldsymbol{\mu}-\varepsilon\mathbf{1}}{\|\boldsymbol{\mu}-\varepsilon\mathbf{1}\|}$ for minimizing the robust error is the same as the optimal solution $\frac{\boldsymbol{\mu}}{\|\boldsymbol{\mu}\|}$ for minimizing the standard error. Otherwise, these two solutions are different, representing a trade-off between robustness and accuracy.*

### A.2   DETAILS FOR THE THEORETICAL ANALYSIS IN SECTION 3

Overall, we would like to design an appropriate synthetic distribution $\widetilde{\mathcal{D}}$ that can help optimize the adversarial classification accuracy in the downstream task. First note that by Bayes rule, the optimal decision boundary for the true distribution $\boldsymbol{x}|y \sim \mathcal{N}(y\boldsymbol{\mu}, \sigma^2\mathbb{I})$ is given by $\boldsymbol{\mu}^{\top}\boldsymbol{x} = 0$. Therefore, we restrict our attention to synthetic data distributions that satisfy the following two conditions:

1. The marginal probability density $p(\tilde{y})$ of the synthetic distribution matches $p(y)$ of the real data distribution well.

2. The conditional probability densities $p(\tilde{\boldsymbol{x}}|\tilde{y} = 1)$ and $p(\tilde{\boldsymbol{x}}|\tilde{y} = -1)$ of the synthetic data distribution are symmetric around the true optimal decision boundary $\boldsymbol{\mu}^{\top}\boldsymbol{x} = 0$.

More specifically, we consider a special case of the synthetic data distribution $\widetilde{\mathcal{D}}_{\mathcal{X}} = 0.5\mathcal{N}(\tilde{\boldsymbol{\mu}}, \sigma^2\mathbb{I}) + 0.5\mathcal{N}(-\tilde{\boldsymbol{\mu}}, \sigma^2\mathbb{I})$.

*Proof of Proposition 1.* We follow the proof strategy in Carmon et al. (2019). Let $b_i$ be the indicator that the $i$-th pseudo-label $\tilde{y}_i$ assigned to $\tilde{x}_i$ is incorrect, so that we have $\tilde{x}_i \sim \mathcal{N}\left((1 - 2b_i)\,\tilde{y}_i\tilde{\mu}, \sigma^2\mathbb{I}\right)$. Let $\gamma := \frac{1}{\tilde{n}}\sum_{i=1}^{\tilde{n}}(1 - 2b_i) \in [-1, 1]$ and $\alpha := \frac{\tilde{n}}{\tilde{n}+n}$. Note that the true data samples $x_i \sim \mathcal{N}\left(y_i\mu, \sigma^2\mathbb{I}\right)$, thus we may write the final estimator as

$$
\hat{\theta}_{\text{final}} = \frac{1}{n+\tilde{n}}\Big(\sum_{j=1}^{\tilde{n}}\tilde{y}_j\tilde{x}_j + \sum_{i=1}^{n}y_i x_i\Big)
$$

$$
= \alpha\gamma\tilde{\mu} + \frac{1}{n+\tilde{n}}\sum_{i=1}^{\tilde{n}}\tilde{y}_i\tilde{\varepsilon}_i + (1-\alpha)\mu + \frac{1}{n+\tilde{n}}\sum_{i=1}^{n}y_i\varepsilon_i
$$

$$
= \alpha\gamma\tilde{\mu} + (1-\alpha)\mu + \frac{1}{n+\tilde{n}}\Big(\sum_{i=1}^{n}y_i\varepsilon_i + \sum_{i=1}^{\tilde{n}}\tilde{y}_i\tilde{\varepsilon}_i\Big),
$$

where $\varepsilon_i, \tilde{\varepsilon}_i \sim \mathcal{N}\left(0, \sigma^2\mathbb{I}\right)$ independent of each other, and the marginal probability density $p(\tilde{y})$ matches $p(y)$ well. Defining $\tilde{\delta} := \hat{\theta}_{\text{final}} - \alpha\gamma\tilde{\mu} - (1-\alpha)\mu$. Note that $\tilde{\delta} \sim \mathcal{N}(0, \frac{1}{n+\tilde{n}}\sigma^2)$.

By (6), we have that the standard error of $f_{\hat{\theta}_{\text{final}}}$ is a non-increasing function of $\frac{\mu^\top\hat{\theta}_{\text{final}}}{\sigma\|\hat{\theta}_{\text{final}}\|}$. Note that when $\tilde{n}$ is large enough, we have the direction of $\hat{\theta}_{\text{final}}$ approach the direction of $\tilde{\mu}$. Therefore, the statement in Case 1 holds as a consequence, and similarly for the robust error according to (7).

The remaining proof on Case 2 and Case 3 is based on a detailed discussion for the squared inverse of the term $\frac{\mu^\top\hat{\theta}_{\text{final}}}{\sigma\|\hat{\theta}_{\text{final}}\|}$:

$$
\frac{\|\hat{\theta}_{\text{final}}\|^2}{(\mu^\top\hat{\theta}_{\text{final}})^2} = \frac{\|\tilde{\delta} + \alpha\gamma\tilde{\mu} + (1-\alpha)\mu\|^2}{(\alpha\gamma\langle\mu,\tilde{\mu}\rangle + \mu^\top\tilde{\delta} + (1-\alpha)\|\mu\|^2)^2}. \tag{8}
$$

Note that the larger the quantity in (8) is, the larger the standard error of $f_{\hat{\theta}_{\text{final}}}$.

Case 2. Assume $\tilde{\mu} = c\mu$. Then we have (8) reduces to:

$$
\frac{\|\hat{\theta}_{\text{final}}\|^2}{(\mu^\top\hat{\theta}_{\text{final}})^2} = \frac{\|\tilde{\delta} + (1 - \alpha + c\gamma\alpha)\mu\|^2}{\Big((1 - \alpha + c\gamma\alpha)\|\mu\|^2 + \mu^\top\tilde{\delta}\Big)^2} \tag{9}
$$

$$
= \frac{1}{\|\mu\|^2} + \frac{\|\tilde{\delta} + (1 - \alpha + c\gamma\alpha)\mu\|^2 - \frac{1}{\|\mu\|^2}\Big((1 - \alpha + c\gamma\alpha)\|\mu\|^2 + \mu^\top\tilde{\delta}\Big)^2}{\Big((1 - \alpha + c\gamma\alpha)\|\mu\|^2 + \mu^\top\tilde{\delta}\Big)^2}
$$

$$
= \frac{1}{\|\mu\|^2} + \frac{\|\tilde{\delta}\|^2 - \frac{1}{\|\mu\|^2}(\mu^\top\tilde{\delta})^2}{\Big((1 - \alpha + c\gamma\alpha)\|\mu\|^2 + \mu^\top\tilde{\delta}\Big)^2}, \tag{10}
$$

which demonstrates that the bigger the $c$ is, the smaller the standard error $\text{err}_{\text{standard}}\left(f_{\hat{\theta}_{\text{final}}}\right)$ is, which verifies the second part of Case 2.

Case 3. Assume $\tilde{\mu} = c(\mu - \varepsilon\mathbf{1}_d)$. Similar to Case 2, we rewrite the term inside the robust error function (7) as:

$$
\frac{\|\hat{\theta}_{\text{final}}\|^2}{\Big((\mu - \varepsilon\mathbf{1}_d)^\top\hat{\theta}_{\text{final}}\Big)^2} = \frac{\|\tilde{\delta} + (1 - \alpha + c\gamma\alpha)(\mu - \varepsilon\mathbf{1}_d)\|^2}{\Big((1 - \alpha + c\gamma\alpha)\|\mu - \varepsilon\mathbf{1}_d\|^2 + (\mu - \varepsilon\mathbf{1}_d)^\top\tilde{\delta}\Big)^2}
$$

$$
= \frac{1}{\|\mu - \varepsilon\mathbf{1}_d\|^2} + \frac{\|\tilde{\delta}\|^2 - \frac{1}{\|\mu - \varepsilon\mathbf{1}_d\|^2}\Big((\mu - \varepsilon\mathbf{1}_d)^\top\tilde{\delta}\Big)^2}{\Big((1 - \alpha + c\gamma\alpha)\|\mu - \varepsilon\mathbf{1}_d\|^2 + (\mu - \varepsilon\mathbf{1}_d)^\top\tilde{\delta}\Big)^2}, \tag{11}
$$

which demonstrates the bigger the $c$ is, the smaller the robust error $\text{err}_{\text{robust}}\left(f_{\hat{\theta}_{\text{final}}}\right)$ is, which proves the second part of Case 3.

$\square$

## B   MORE SIMULATION RESULTS UNDER GAUSSIAN SETTING IN SECTION 3

In this section, we present more detailed simulation results under the Gaussian setting in Section 3 to demonstrate different scenarios in Proposition 1. Table 7 and Table 8 show the clean and robust accuracy learned on synthetic distribution $\tilde{\mu} = c\mu$ with different angles between $\mu$ and $\epsilon\mathbf{1}_d$. Table 10 shows the clean and robust accuracy learned on synthetic distribution $\tilde{\mu} = c(\mu - \varepsilon\mathbf{1}_d)$ with different angles between $\mu$ and $\epsilon\mathbf{1}_d$. Recall that $\mu$ is (one of) the optimal linear classifier that maximize the clean accuracy under the true distribution considered in Section 3, similarly $\mu - \epsilon\mathbf{1}_d$ is the optimal solution for robust accuracy. Therefore, different angles between $\mu$ and $\epsilon\mathbf{1}_d$ represent different trade-offs between the clean and robust accuracy. For example, when the angle between $\mu$ and $\epsilon\mathbf{1}_d$ is 0 degrees, i.e., $\mu = c\mathbf{1}_d$, we have that the optimal solution for clean accuracy and robust accuracy are the same. In most cases, the classifier learned from the synthetic distribution that is most separable achieves better performance even than the iid samples, which verifies Proposition 1.

Table 7: The clean and robust accuracy learned on synthetic distribution $\tilde{\mu} = c\mu$ when $d = 2$ and the angle between $\mu$ and $\epsilon$ is 0 degrees and 90 degrees. "Real" denotes the real data distribution, and $n$ denotes the number of data from the real distribution, while we use "$c$" to denote different synthetic distributions and use $\tilde{n}$ to denote the number of synthetic data. The results and the standard deviation in the bracket are the results of 50 independent trials.

|  |  | 0 degree | | 90 degree | |
|---|---|---|---|---|---|
|  |  | acc (std) | rob acc (std) | acc (std) | rob acc (std) |
| Real | $n = 10$ | 0.9201 (0.0012) | **0.7593** (0.0020) | 0.9171 (0.0046) | 0.7552 (0.0040) |
|  | $n = 20$ | 0.9204 (0.0007) | **0.7598** (0.0016) | 0.9186 (0.0017) | 0.7563 (0.0012) |
|  | $n = 50$ | 0.9206 (0.0004) | **0.7605** (0.0007) | 0.9196 (0.0009) | 0.7566 (0.0006) |
|  | $n = 100$ | 0.9205 (0.0004) | **0.7608** (0.0006) | 0.9199 (0.0006) | 0.7565 (0.0007) |
| $c = 0.5$ | $\tilde{n} = 10$ | 0.9159 (0.0099 ) | 0.7541 (0.0096) | 0.9104 (0.0121) | 0.7492 (0.0122) |
|  | $\tilde{n} = 20$ | 0.9179 (0.0047) | 0.7562 (0.0050) | 0.9161 (0.0052) | 0.7546 (0.0054) |
|  | $\tilde{n} = 50$ | 0.9200 (0.0023) | 0.7586 (0.0024) | 0.9183 (0.0022) | 0.7570 (0.0022) |
|  | $\tilde{n} = 100$ | 0.9213 (0.0011) | 0.7601 (0.0009) | 0.9193 (0.0012) | 0.7576 (0.0010) |
| $c = 1$ | $\tilde{n} = 10$ | 0.9133 (0.0066) | 0.7502 (0.0061) | 0.9161 (0.0048) | **0.7598** (0.0048) |
|  | $\tilde{n} = 20$ | 0.9155 (0.0020) | 0.7516 (0.0019) | 0.9180 (0.0017) | **0.7612** (0.0020) |
|  | $\tilde{n} = 50$ | 0.9161 (0.0009) | 0.7525 (0.0006) | 0.9186 (0.0010) | **0.7620** (0.0006) |
|  | $\tilde{n} = 100$ | 0.9165 (0.0005) | 0.7528 (0.0006) | 0.9189 (0.0005) | **0.7622** (0.0003) |
| $c = 1.5$ | $\tilde{n} = 10$ | **0.9209** (0.0038) | 0.7523 (0.0025) | **0.9221** (0.0017) | 0.7583 (0.0015) |
|  | $\tilde{n} = 20$ | **0.9228** (0.0010) | 0.7536 (0.0006 ) | **0.9226** (0.0013) | 0.7588 (0.0013) |
|  | $\tilde{n} = 50$ | **0.9229** (0.0008) | 0.7538 (0.0005) | **0.9232** (0.0005) | 0.7594 (0.0006) |
|  | $\tilde{n} = 100$ | **0.9232** (0.0003) | 0.7538 (0.0005) | **0.9233** (0.0005) | 0.7595 (0.0005) |

Table 8: The clean and robust accuracy learned on synthetic distribution $\tilde{\mu} = c\mu$ when $d = 2$ and the angle between $\mu$ and $\epsilon$ is 30 degrees and 60 degrees. "Real" denotes the real data distribution, and $n$ denotes the number of data from the real distribution, while we use "$c$" to denote different synthetic distributions and use $\tilde{n}$ to denote the number of synthetic data. The results and the standard deviation in the bracket are the results of 50 independent trials.

| | | 30 degree | | 60 degree | |
| --- | --- | --- | --- | --- | --- |
| | | acc (std) | rob acc (std) | acc (std) | rob acc (std) |
| Real | $n = 10$ | 0.8307 (0.0123) | 0.6343 (0.0283) | 0.8348 (0.0117) | 0.6378 (0.0293) |
| | $n = 20$ | 0.8353 (0.0055) | 0.6404 (0.0234) | 0.8391 (0.005) | 0.6433 (0.0222) |
| | $n = 50$ | 0.8371 (0.0022) | 0.6450 (0.0168) | 0.8410 (0.0017) | 0.6494 (0.0134) |
| | $n = 100$ | 0.8385 (0.0010) | 0.6461 (0.0097) | 0.8413 (0.0013) | 0.6522 (0.0102) |
| $c = 0.5$ | $\tilde{n} = 10$ | 0.8265 (0.0184 ) | 0.6282 (0.0418 ) | 0.8338 (0.0132 ) | 0.6303 (0.0335 ) |
| | $\tilde{n} = 20$ | 0.8299 (0.0129) | 0.6352 (0.0325) | 0.8365 (0.0132) | 0.6393 (0.0316) |
| | $\tilde{n} = 50$ | 0.8372 (0.0046) | 0.6483 (0.0215) | 0.8414 (0.0034) | 0.6489 (0.0199) |
| | $\tilde{n} = 100$ | 0.8402 (0.0015) | 0.6466 (0.0110) | 0.8431 (0.0012) | 0.6510 (0.0135) |
| $c = 1$ | $\tilde{n} = 10$ | 0.8383 (0.0158) | 0.6439 (0.0319) | **0.8377** (0.0074) | 0.6396 (0.0267) |
| | $\tilde{n} = 20$ | 0.8425 (0.0060) | 0.6480 (0.0218) | **0.8416** (0.0034) | **0.6513** (0.0178) |
| | $\tilde{n} = 50$ | 0.8455 (0.0023) | 0.6553 (0.0128) | **0.8432** (0.0020) | **0.6503** (0.0122) |
| | $\tilde{n} = 100$ | 0.8457 (0.0021) | 0.6535 (0.0100) | **0.8435** (0.0014) | **0.65011** (0.0096) |
| $c = 1.5$ | $\tilde{n} = 10$ | **0.8431** (0.0045) | **0.6542** (0.0173) | 0.8368 (0.0073) | **0.6446** (0.0213) |
| | $\tilde{n} = 20$ | **0.8447** (0.0021) | **0.6542** (0.0142) | 0.8393 (0.0022) | 0.6479 (0.0150) |
| | $\tilde{n} = 50$ | **0.8455** (0.0006) | **0.6556** (0.0082) | 0.8404 (0.0005) | 0.6488 (0.0089) |
| | $\tilde{n} = 100$ | **0.8457** (0.0004) | **0.6547** (0.0057) | 0.8404 (0.0007) | 0.6486 (0.0082) |

Table 9: The clean and robust accuracy learned on synthetic distribution $\tilde{\mu} = c\mu$ when $d = 100$ and the angle between $\mu$ and $\epsilon$ is 0 degrees. "Real" denotes the real data distribution, and $n$ denotes the number of data from the real distribution, while we use "$c$" to denote different synthetic distributions and use $\tilde{n}$ to denote the number of synthetic data. The results and the standard deviation in the bracket are the results of 50 independent trials.

| | | acc (std) | rob acc (std) |
| --- | --- | --- | --- |
| Real | $n = 10$ | 0.9023 (0.0192) | 0.6843 (0.0359) |
| | $n = 20$ | 0.9341 (0.0128) | 0.7519 (0.0267) |
| | $n = 50$ | 0.9599 (0.0028) | 0.8078 (0.0061) |
| | $n = 100$ | 0.9682 (0.0014) | 0.8239 (0.0028) |
| $c = 0.5$ | $\tilde{n} = 10$ | 0.7562 (0.0564) | 0.4611 (0.0694) |
| | $\tilde{n} = 20$ | 0.8566 (0.0307) | 0.6047 (0.0491) |
| | $\tilde{n} = 50$ | 0.9261 (0.0117) | 0.7328 (0.0227) |
| | $\tilde{n} = 100$ | 0.9505 (0.0047) | 0.7848 (0.0111) |
| $c = 1$ | $\tilde{n} = 10$ | 0.8866 (0.0273) | 0.6557 (0.0487) |
| | $\tilde{n} = 20$ | 0.9371 (0.0091) | 0.7555 (0.0201) |
| | $\tilde{n} = 50$ | 0.9620 (0.0028) | 0.8085 (0.0060) |
| | $\tilde{n} = 100$ | 0.9695 (0.0012) | 0.8239 (0.0031) |
| $c = 1.5$ | $\tilde{n} = 10$ | **0.9400** (0.0100) | **0.7603** (0.0233) |
| | $\tilde{n} = 20$ | **0.9591** (0.0037) | **0.8031** (0.0080) |
| | $\tilde{n} = 50$ | **0.9710** (0.0013) | **0.8280** (0.0028) |
| | $\tilde{n} = 100$ | **0.9743** (0.0008) | **0.8343** (0.0011) |

Table 10: The clean and robust accuracy learned on synthetic distribution $\tilde{\boldsymbol{\mu}} = c(\boldsymbol{\mu} - \varepsilon\mathbf{1}_d)$ when $d = 2$ and the angle between $\boldsymbol{\mu}$ and $\epsilon$ is 30 degrees and 60 degrees. "Real" denotes the real data distribution, and $n$ denotes the number of data from the real distribution, while we use "$c$" to denote different synthetic distributions and use $\tilde{n}$ to denote the number of synthetic data. The results and the standard deviation in the bracket are the results of 50 independent trials.

|  |  | 30 degree | | 60 degree | |
|  |  | acc (std) | rob acc (std) | acc (std) | rob acc (std) |
|---|---|---|---|---|---|
| Real | $n = 10$ | 0.9152 (0.0049) | 0.7633 (0.0111) | **0.9211** (0.0034) | 0.7702 (0.0094) |
|  | $n = 20$ | 0.9170 (0.0030) | 0.7642 (0.0075) | **0.9225** (0.0020) | 0.7714 (0.0068) |
|  | $n = 50$ | 0.9183 (0.0009) | 0.7653 (0.0040) | **0.9232** (0.0011) | 0.7711 (0.0050) |
|  | $n = 100$ | 0.9185 (0.0006) | 0.7658 (0.0027) | **0.9235** (0.0009) | 0.7724 (0.0027) |
| $c = 0.5$ | $\tilde{n} = 10$ | 0.9089 (0.0183) | 0.7563 (0.0310) | 0.9111 (0.0114) | 0.7638 (0.0172) |
|  | $\tilde{n} = 20$ | 0.9144 (0.0068) | 0.7659 (0.0107) | 0.9138 (0.0068) | 0.7694 (0.0066) |
|  | $\tilde{n} = 50$ | 0.9174 (0.0029) | 0.7680 (0.0068) | 0.9161 (0.0038) | 0.7714 (0.0033) |
|  | $\tilde{n} = 100$ | 0.9183 (0.0016) | 0.7681 (0.0053) | 0.9165 (0.0031) | 0.7727 (0.0014) |
| $c = 1$ | $\tilde{n} = 10$ | 0.9135 (0.0116) | 0.7642 (0.0194) | 0.9069 (0.0111) | 0.7677 (0.0109) |
|  | $\tilde{n} = 20$ | 0.9178 (0.0046) | 0.7710 (0.0073) | 0.9042 (0.0098) | 0.7676 (0.0072) |
|  | $\tilde{n} = 50$ | 0.9183 (0.0042) | 0.7728 (0.0042) | 0.9073 (0.0047) | 0.7702 (0.0017) |
|  | $\tilde{n} = 100$ | 0.9196 (0.0017) | 0.7733 (0.0036) | 0.9059 (0.0039) | 0.7698 (0.0016) |
| $c = 1.5$ | $\tilde{n} = 10$ | **0.9181** (0.0079) | **0.7747** (0.0104) | 0.9034 (0.0079) | **0.7704** (0.0053) |
|  | $\tilde{n} = 20$ | **0.9209** (0.0053) | **0.7770** (0.0052) | 0.9077 (0.0059) | **0.7716** (0.0056) |
|  | $\tilde{n} = 50$ | **0.9218** (0.0029) | **0.7788** (0.0028) | 0.9073 (0.0030) | **0.7722** (0.0014) |
|  | $\tilde{n} = 100$ | **0.9222** (0.0017) | **0.7793** (0.0023) | 0.9077 (0.0024) | **0.7729** (0.0011) |

# C  THE DETAILED CONSTRUCTION OF THE CONTRASTIVE LOSS

In this section, we first give a detailed description of several possible ways to design contrastive loss, especially in constructing positive and negative pairs. Then, we give a visualization of the synthetic data distributions generated under different contrastive losses.

## C.1  POSITIVE AND NEGATIVE PAIR SELECTION STRATEGY.

In this subsection, we give several possible ways to construct positive and negative pairs.

1. Vanilla version: Using all the samples in the minibatch is the common strategy for contrastive learning. In the diffusion process, since for each time step $t$, we want to distinguish each image from other images in the minibatch at the same time step, a straight-forward strategy is to use all the samples in the minibatch other than $x_t^i$ at time step $t$ to be the negative pairs. For the positive pairs, we can simply adopt $x_{t+1}^i$ to be the positive pairs rather than augmentation of $x_t^i$.

2. Real data as positive pairs: A possible improvement upon the vanilla version is considering we aim to generate images similar to real data. Therefore, we can directly adopt the real data as the positive pairs.

3. Real data as negative pairs: Another improvement upon the vanilla version is considering the other images in time step $t$ in the minibatch is not as high quality as the real data. Therefore, we can directly adopt the real data as the negative pairs.

4. Class conditional version: When we use conditional diffusion, and the class label of $x_t$ in the minibatch is available, a further improvement can be adopted is to use all the samples with different class label $y$ in the minibatch at time step $t$ to be the negative pairs.

## C.2  VISUALIZATION OF THE SYNTHETIC DATA DISTRIBUTION GENERATED BY DIFFERENT DESIGNS OF THE CONTRASTIVE LOSS

In this subsection, we demonstrate the synthetic distributions generated by different designs of the contrastive loss mentioned in Section C.1 on the Gaussian setting mentioned in Section 3.1. Figure 3 shows the synthetic distribution generated by using $\mathcal{N}(\mathbf{0}, \mathbb{I})$ as initialization, while Figure 4 shows the synthetic distribution generated by using $\mathcal{N}(\mathbf{0}, 4\mathbb{I})$ as initialization. In all figures, all of the contrastive loss except for conditional hard negative mining form a circle within each class, which means these algorithms cannot explicitly distinguish the data within the same class and thus maximize the distance within each class, while the guidance from conditional hard negative mining can generate samples that are more distinguishable.

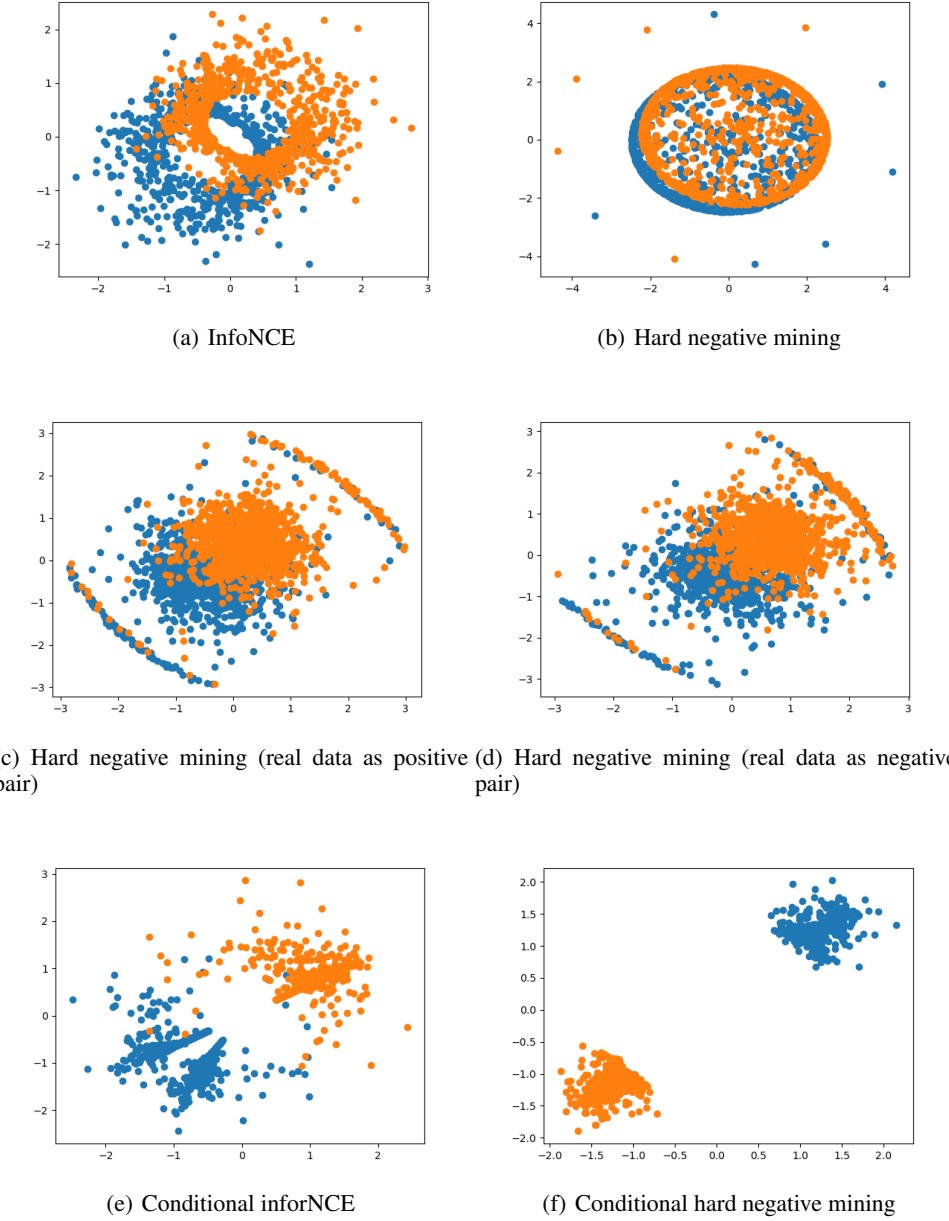

(a) InfoNCE

(b) Hard negative mining

(c) Hard negative mining (real data as positive pair)

(d) Hard negative mining (real data as negative pair)

(e) Conditional inforNCE

(f) Conditional hard negative mining

Figure 3: A comparison of the synthetic distribution guided by different contrastive loss with initialization $\mathcal{N}(\mathbf{0}, \mathbb{I})$. Real data as positive pair means using the mixture of oracle distribution $\mathcal{N}(\pm\mathbf{1}, \mathbb{I})$ and the data in the same batch as negative pair, while real data as negative pair means using the data in the same batch as positive pair and using the mixture of oracle distribution as negative pair.

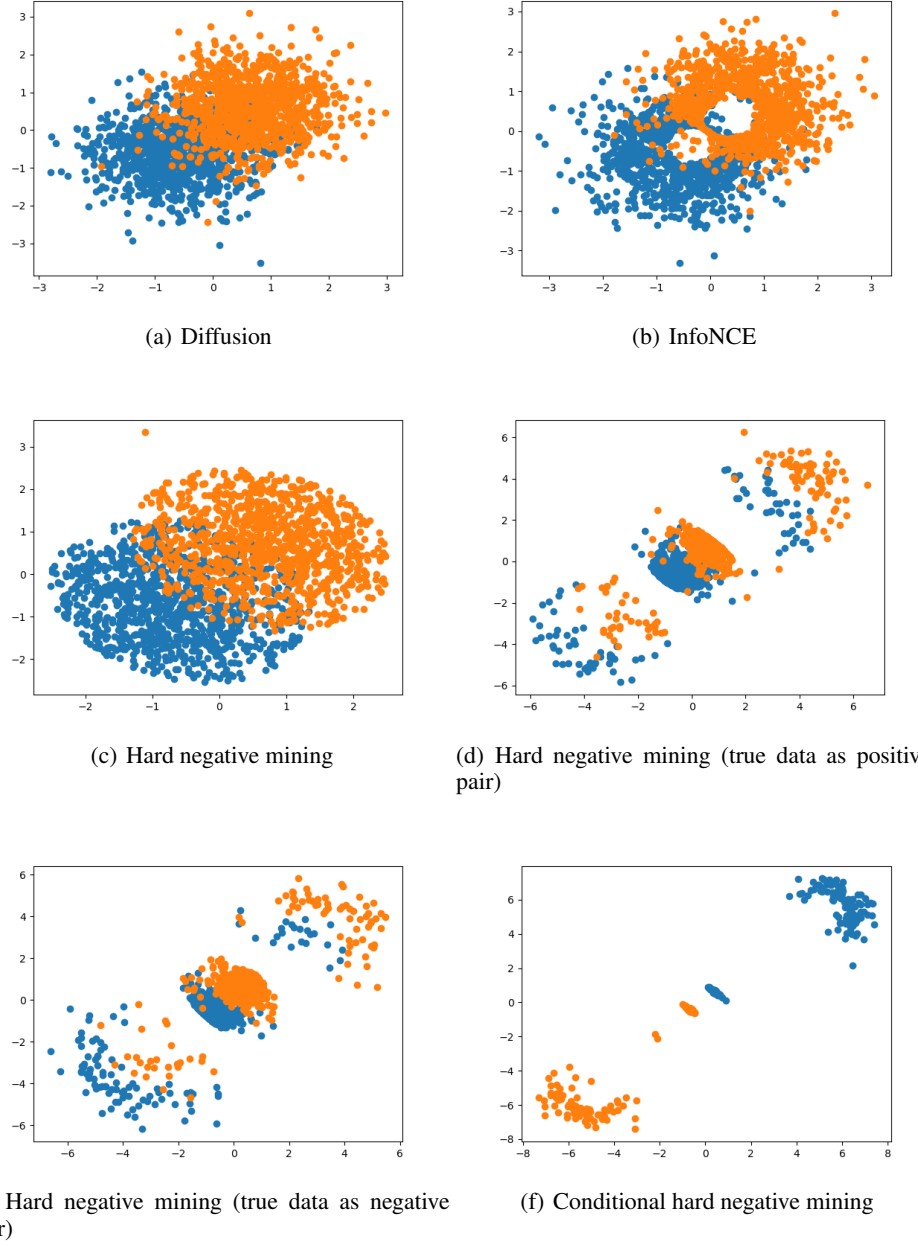

(a) Diffusion

(b) InfoNCE

(c) Hard negative mining

(d) Hard negative mining (true data as positive pair)

(e) Hard negative mining (true data as negative pair)

(f) Conditional hard negative mining

Figure 4: A comparison of the synthetic distribution guided by different contrastive loss with initialization $\mathcal{N}(\mathbf{0}, 4\mathbb{I})$.

# D THE EXPERIMENTAL RESULTS IN THE REAL-WORLD SETTING

## D.1 EXPERIMENTAL SETUP FOR CIFAR-10 DATASET

We describe the pipeline of synthetic data generation for adversarial robustness and a corresponding setting for the CIFAR-10 dataset in this subsection.

**Dataset.** CIFAR-10 dataset Krizhevsky (2009) contains 50K 32x32 color training images in 10 classes and 10K images for testing.

**Overall training pipeline** We follow the same training pipeline as Gowal et al. (2021), i.e., synthesizing data by using the diffusion model, assigning pseudo-label for synthetic data and aggregate the original data and the synthetic data for adversarial training. We give a careful explanation of these three components as follow.

**Synthetic data generation by the diffusion model.** Considering the advantage of DDIM on generation speed, we base on the official implementation of the DDIM model (Song et al., 2021a) and add the guidance of the contrastive loss. We generate images with 200 steps with batchsize=512, and use the quadratic version of sub-sequence selection [3]. For the guidance of the contrastive loss, we try different designs of the contrastive loss mentioned in Section 4.2. We set the temperature $\tau = 0.1$ and the strength of guidance of the contrastive loss $\lambda = 20k$ in the InfoNCE loss, while $\tau = 10$, the strength of guidance of the contrastive loss $\lambda = 100k$, the probability of the same class in the minibatch $\tau^+ = 0.1$ and the hardness of negative mining $\beta = 1$ in hard negative mining loss. These corresponding hyperparameters are chosen based on some preliminary experiments on image generation. The detailed ablation studies can be found in Section 5.2. Moreover, we also delve into the representation used by contrastive loss. The default setting is to use the pre-trained Wide ResNet-28-10 model Gowal et al. (2021) to get the representation for applying the contrastive loss, which is named as (without embedding) in Section 5.2. A further improvement is to apply a 2-layer feed-forward neural network to encode the representation after the pre-trained model, which is named as (with embedding). The advantage of the latter design is we can adopt the contrastive loss to optimize the encoding network rather than a fixed encoder.

**LaNet for assigning pseudo-label.** Since the DDIM is an unconditional generator, we need to assign the pseudo-label to the generated sample. We follow the same choice adopted by Sehwag et al. (2022), i.e., using state-of-the-art LaNet Wang et al. (2019) network for assigning the pseudo-label for the synthetic data.

**Adversarial Training.** We follow the same setting as Gowal et al. (2021), i.e., we use Wide ResNet-28-10 Zagoruyko & Komodakis (2016) with Swish activation function Hendrycks & Gimpel (2016), adopt stochastic weight averaging Izmailov et al. (2018) with decay rate 0.995 and use TRADES Zhang et al. (2019) with 10 Projected Gradient Descent steps and $\varepsilon_\infty = 8/255$ for 400 epochs with batch size 1024[4].

**Evaluation setup** We follow the same training pipeline as Gowal et al. (2021), i.e., for each trained model, we adopt either AUTOATTACK Croce & Hein (2020) or AA+MT Gowal et al. (2020) with $\epsilon_\infty = 8/255$ and report the worst accuracy as robust accuracy.

## D.2 EXPERIMENTAL SETUP FOR TRAFFIC SIGNS DATASET

We describe the pipeline of synthetic data generation for adversarial robustness and a corresponding setting for the Traffic Signs dataset in this subsection.

---

[3] We refer to Appendix D.2 for a detailed explanation of the quadratic version.

[4] For Table 5 in the ablation studies subsection, we use batch size with 256.

**Dataset.** Traffic Signs dataset Houben et al. (2013) contains 39252 training images in 43 classes and 12629 images for testing, and the image sizes vary between 15x15 to 250x250 pixels.

**Synthetic data generation by the diffusion model.** To utilize the pre-trained diffusion model [5], we use a conditional DDPM for generating samples for Traffic Signs dataset. We adopt the hard negative mining loss with $\tau = 10$, the strength of guidance of the contrastive loss $\lambda = 5k$, the probability of the same class in the minibatch $\tau^+ = 0.1$ and the hardness of negative mining $\beta = 1$. We also use the pre-trained Wide ResNet-28-10 model to get the representation for applying the contrastive loss and use a 2-layer feed-forward neural network to encode the representation after the pre-trained model.

**Adversarial Training.** We follow the same setting as the CIFAR-10 dataset, except the training epochs are reduced to 50. We also extend the training epochs to 400 but do not find significant improvement.

### D.3 THE DETAILED EXPLANATION OF THE DATA SELECTION METHODS

Below we summarize different data selection methods:

- DDIM (Separability): We adopt the separability of the data as a criterion to make the selection of the data generated by vanilla DDIM. For each data, we use a pre-trained WRN-28-10 model to encode them into the embedding space. Then, we compute the L2 distance between each sample and the centroid of all classes (which is easily computed as the mean of all samples in this class) and add them together. To select a subset of samples that are most distinguishable, we choose the top K samples that have the smallest distance in each class.

- Contrastive-DP (Gradient norm): We use the gradient norm with respect to a pre-trained WRN-28-10 model as a criterion to make the selection on the data generated by Contrastive-DP. The larger the gradient norm is, the more informative the sample is for learning a downstream model. Therefore, we select the top $K$ samples that have the largest gradient norm in each class.

- Contrastive-DP (Gradient norm-rob): Similar to Contrastive-DP (Gradient norm), we use the gradient norm of the robust loss rather than standard classification loss as a criterion to make the selection on the data generated by Contrastive-DP. Therefore, we select the top $K$ samples that have the largest gradient norm in each class.

- Contrastive-DP (Entropy): We use the entropy of each sample with respect to LaNet as a criterion to make the selection on the data generated by Contrastive-DP. The smaller the entropy is, the higher likelihood this image has good quality. Therefore, we select the top $K$ samples that have the smallest entropy in each class.

### D.4 COMPARISON OF THE IMAGE GENERATED BY CONTRASTIVE-GP WITH THE VANILLA DDIM

In this subsection, we visualize the image generated by Contrastive-GP and the vanilla DDIM on the CIFAR-10 dataset. We find the guidance of the contrastive loss changes the category of the synthetic images or makes the synthetic images realistic (colorful).

### D.5 T-SNE ON THE CLASSIFIER LEARNED ON SYNTHETIC DATA.

In this subsection, we visualize the t-SNE of the finial classifier learned on different synthetic data. We find with the guidance of the contrastive loss, the finial classifier learns a better representation that makes the feature of the images from different classes more separable than the finial classifier learned on the images generated by the vanilla DDIM.

---

[5] https://github.com/VSehwag/minimal-diffusion

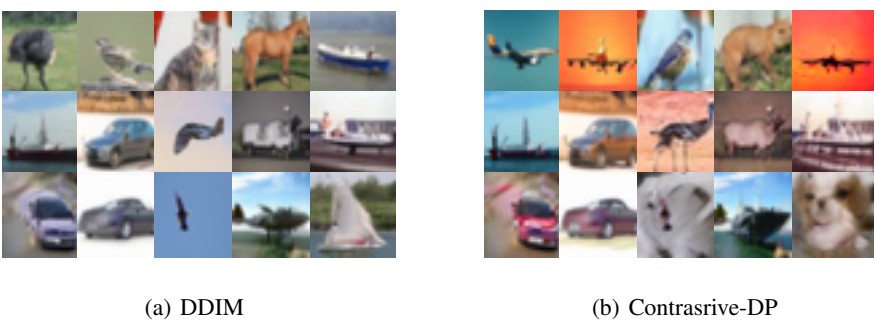

(a) DDIM                              (b) Contrasrive-DP

Figure 5: Comparison of the Contrastive-GP with the vanilla DDIM. The image in the same position on subfigures (a) and (b) has the same initialization. With the guidance of the contrastive loss, the category of the synthetic images changes, or the synthetic images become more realistic (colorful), which demonstrates the effectiveness of our Contrastive-DP algorithm.

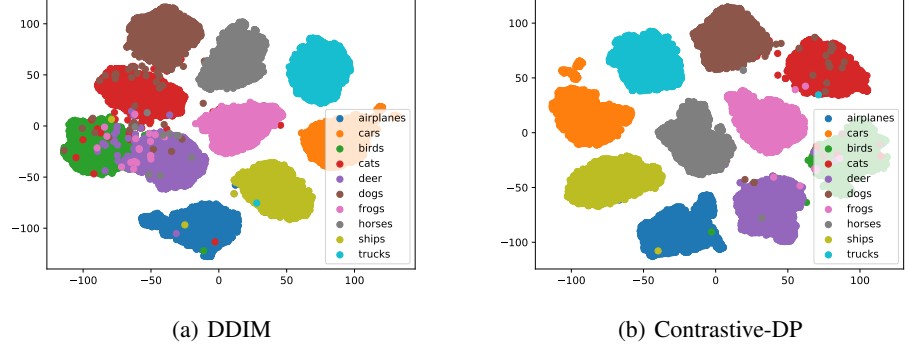

(a) DDIM                              (b) Contrastive-DP

Figure 6: A comparison of the T-SNE of the finial classifier learned on different synthetic data on the CIFAR-10 dataset.

