# OpenReview forum: "Improving Adversarial Robustness by Contrastive Guided Diffusion Process"
_ICLR.cc/2023/Conference — Submitted to ICLR 2023_

### Official Review · Reviewer_H9p7 · 2022-10-13

**Confidence:** 4
**Correctness:** 3
**Technical Novelty And Significance:** 2
**Empirical Novelty And Significance:** 2
**Recommendation:** 3

**Clarity, Quality, Novelty And Reproducibility:**

The proposed technique is not clearly stated, and the novelty is limited. The results should be able to reproduce with an additional description of the missing details in the proposed technique. The overall quality is below the bar.

**Details Of Ethics Concerns:**

This paper studies the security of DNN and ML models. It would safe to include an ethical statement. The authors could discuss how the proposed technique could be used by attackers and defenders.

**Strength And Weaknesses:**

Strengths
+ The paper provides a clear description of the background.
+ The theoretical analysis part is clearly described.

Weaknesses:
- Regarding the theoretical analysis, as mentioned above, I appreciate that the analysis part is clear. However, I have the following two questions: (1) For the simulation test, the authors mainly focus on the case $\hat{\mu}=c\mu$. The case that improves the clean accuracy. I am wondering why not verify the other case $\hat{\mu}=c(\mu - \epsilon 1_d)$ that improves the robust accuracy. (2) It would be great if the authors could provide more justification for the conclusion drawn from the theoretical analysis: "The synthetic data can help improve the classification task especially when the representation of different classes is more distinguishable in the synthetic distribution." From the analysis, the most obvious conclusion one could draw is under what situation the clean accuracy is the best ( $\hat{\mu}=c\mu$), and under what situation, robust accuracy is the best. This situation is based on the difference between the synthetic distribution and the original distribution. It is not that obvious between the connection between the above situations to the difference between the representations of different classes in the synthetic data.

- The proposed technique is not entirely clear to me. The proposed data augmentation is clear. But the overall training process is not clear. I am not sure what the procedure would be like. It either generates data using the proposed method and uses the augmented dataset to train the model or generates data using the proposed method and applies adversarial training on the augmented data, or others. My understanding is the second one. It would be great if the authors could clearly describe the overall procedure. If my understanding is correct, it brings a new question, whether the robustness comes from data augmentation or adversarial training. The paper does not justify this.

- The evaluation setup is not clear enough, mainly how to generate attacks. My understanding is training a model using the proposed method and then generating attacks based on the trained model. Another way is to generate attacks on a vanilla model and use those adversarial samples to test models trained by the proposed method. It would be great if the authors could make this setup clear. The improvement over DDPM and DDIM is marginal (BTW, what is the reason for using DDIM for CIFAR-10 and DDPM for Traffic sign data). I would also suggest reporting the mean and standard errors of multiple runs to show the improvement is not because of randomness.

**Summary Of The Paper:**

This paper proposes a diffusion model-based data augmentation method. The proposed method combines the contrastive loss with the diffusion model generation process to generate data with a large diversity between different classes. It shows that using the proposed method generates data and trains a classifier, the trained classifier is more robust to adversarial attacks than DDIM, the SOTA diffusion model.

**Summary Of The Review:**

This paper proposes a new defense to train adversarially robust classifiers. As mentioned above, the descriptions of the proposed method are not entirely clear, and the technical contribution is not that significant. Besides, the empirical improvement is marginal and is not well justified.

---

> ### Author Response · Authors · 2022-11-18
> **For Reviewer H9p7**
>
> We thank the reviewer for the comments and suggestions. We appreciate the time you spent on the paper. Below we address the concerns and comments that you have provided.
>
> **Q**: *Why not verify the other case $\hat{\mu}=c(\mu - \epsilon 1_d)$ that improves the robust accuracy?*
>
> **A**: Thank you very much for this suggestion. We conduct a detailed comparison and add the corresponding result in Appendix B, i.e., Table 11. The new results now still validate our theoretical findings in Proposition 1 (Case 3).
>
> **Q**: *The overall training process is not clear.*
> **A**: Thank you very much for pointing this out. We clarify the training pipeline in Appendix D.1, i.e., we follow the same training pipeline as the state-of-the-art work (Gowal et al. 2021). Your second understanding is correct. However, all of the methods (baseline, DDIM, and ours) in Table 2 adopt adversarial training. The better performance achieved by our method against DDIM demonstrates the improvement comes from data augmentation.
>
> **Q**: *The evaluation setup is not clear enough, mainly how to generate attacks.*
> **A**: Your first understanding is correct, which is the standard evaluation for adversarial robustness. We clarify the evaluation setup in Appendix D.1, i.e., we follow the same evaluation setup as the state-of-the-art work (Gowal et al. 2021).
>
> **Q**: *What is the reason for using DDIM for CIFAR-10 and DDPM for Traffic sign data?*
> **A**: We generate the data mainly based on the retrained model given in Ho et. al. (2021) and \url{https://github.com/VSehwag/minimal-diffusion\#how-useful-is-synthetic-data-from-diffusion-models-}, which is the default setting of each repository.
>
> Reference: Ho, J., Jain, A., & Abbeel, P. (2020). Denoising Diffusion Probabilistic Models. NIPS.
>
> **Q**: *I would also suggest reporting the mean and standard errors of multiple runs to show the improvement is not because of randomness.*
>
> **A**: Thank you very much for this suggestion. The numerical experiments were conducted on 4 NVIDIA A100 Tensor Core GPUs, and the running time for DDIM and contrastive-DP is comparable (about two days). Based on our empirical observations, the results were consistent across multiple runs, and we were not able to include the standard error resulting from multiple runs due to limited computing resources. We have reported the results on Traffic Signs dataset for three runs on the revised version (Table 4), and We will add these in the future revised version.

---

### Official Review · Reviewer_96gf · 2022-10-19

**Confidence:** 3
**Correctness:** 3
**Technical Novelty And Significance:** 2
**Empirical Novelty And Significance:** 3
**Recommendation:** 3

**Clarity, Quality, Novelty And Reproducibility:**

Clarity: The paper is clearly written and easy to follow.

Quality: The paper clearly states the hypothesis under analysis, and provides a simplified framework from which design decisions used in practical cases are inspired.

Novelty: The approach employed to guide data generation is novel to my knowledge.

Reproducibility: While authors did include some implementation details in the text, it seems to me that it would be difficult to reproduce results without access to code.



**Strength And Weaknesses:**

Pros:

+The paper tackles relevant problems from a practical perspective: how to best leverage synthetic data in order to obtain robust classifiers.

+The practical approach is motivated from findings obtained formally in a simplified setting, rather than solely based on intuition.

+The approach used to sample synthetic data might be more generally applicable in settings other than the ones covered in the paper.

Cons:

-It's unclear whether observed improvements are due to the properties of the synthetic data or the simple fact the training sample became bigger. I would suggest an experiment where you train models on a subsample of the training data (say half of it) supplemented by synthetic data up to the same size as the original sample. Comparing this model with the one training on the full training sample would help control for the contribution of the sample size.

-Results aren't contextualized with past work in that numbers reported in previous work are not reported. In particular, baselines results without additional data in table 2 seem weak (entries 7 and 11 in this leaderboard seem so use the same architecture and threat model: https://robustbench.github.io/#div_cifar10_Linf_heading).

-It's also unclear to what extent the results observed in the simplified setting transfer to real data. It seems to me the results are a direct consequence of the choice of model class, so it seems the optimal synthetic data source is the one that introduces points lying far from the decision boundary. I wonder how conclusions would change had a different model class been chosen, e.g., SVMs.

**Summary Of The Paper:**

This paper studies the setting where supplemental synthetic data is used to augment the training sample and improve generalization. Specifically, authors focus on the adversarial setting where prediction errors consider worst case perturbations within a neighbourhood of test points. In particular, in a simplified setting, authors verified necessary properties of the synthetic data source in order fo it to yield performance gains: synthetic class conditionals must be far from their real counterparts, while still being separable by the same the decision boundary as the original data. They then use this insight and define image generation procedures that generate samples that, while still being perceptually close to the data distribution, it is as far as possible from it. Empirical assessment shows improvements when using the proposed "guided" generation approach combined with adversarial training.

**Summary Of The Review:**

The paper tackles a relevant problem and proposes practical approaches to guide generative processes towards samples that do not trivially match the training data. However, it's unclear to me whether the observed improvements are due to simply increasing the sample size or to the specific properties of the generated data. Moreover, I'm not confident that results on CIFAR-10 are at least on par with recent work under similar settings.

---

> ### Author Response · Authors · 2022-11-18
> **For Reviewer 96gf**
>
> We thank the reviewer for the comments and suggestions. We appreciate the time you spent on the paper. Below we address the concerns and comments that you have provided.
>
> **Q**: *It's unclear whether observed improvements are due to the properties of the synthetic data or the simple fact the training sample became bigger. I would suggest an experiment where you train models on a subsample of the training data (say half of it) supplemented by synthetic data up to the same size as the original sample.*
>
> **A**: Thank you very much for your good suggestion. In terms of verifying the effectiveness of the proposed generation method, our experimental setting is a fair comparison: we compare the performance of the model trained on the additional image generated by the Contrastive Guided Diffusion model and vanilla diffusion model (DDIM) and demonstrate the effectiveness of our methods by Table 2 and 3. The improvement is significant for a small data regime (50K) and not significant for a large data regime (1M), which is reasonable as the robust accuracy improves when the number of training data is increased. And we would like to emphasize that the significant improvement for a small data regime (50K) also shows that the proposed method tends to be more robust in the limited-data-sample regime, giving a lot of potential for real applications where the labeled real data could be limited.
>
> **Q**: *Results aren't contextualized with past work in that numbers reported in previous work are not reported. In particular, baselines results without additional data in table 2 seem weak (entries 7 and 11 in this leaderboard seem so use the same architecture and threat model: https://robustbench.github.io/#div_cifar10_Linf_heading).*
>
> **A**: We would like to clarify a misunderstanding on the leaderboard. Firstly, entries 7 and 11 actually use additional data. Entries 7 uses synthetic data generated by DDPM, and entries 11 use the data from the 80 Million Tiny Images dataset. (In the leaderboard, they regard using  synthetic data as "without additional data", and using the 80 Million Tiny Images dataset as "with additional data", which is different from the meaning in our Table 2.) Secondly, the state-of-the-art performance currently is achieved by Gowal et al. (2021). They utilize 100M synthetic data, which takes more than three months to generate by using 4 NVIDIA A100 Tensor Core GPUs. This is computationally prohibited in our case. To make sure a fair comparison is conducted, we rerun all the experiments on $50K/200K/1M$ settings for comparison (As mentioned in our main paper, i.e., the footnote on page 8, since the Pytorch Implementation of Gowal et al. (2021) is not open source, we utilize the best unofficial implementation to reconduct all the experiments for a fair comparison).
>
> **Q**: *It's also unclear to what extent the results observed in the simplified setting transfer to real data ... I wonder how conclusions would change had a different model class been chosen, e.g., SVMs.*
>
> **A**: Thank you very much for this question. Firstly, we would like to emphasize that the Gaussian mixture setting is widely used in theoretical works, e.g., (Carmon et al., 2019), (Deng et al., 2021), and (Sorscher et. al., 2022), in order to provide theoretical insights for the downstream adversarial learning tasks. And the real data distribution is unknown and complicated, making it hard to be analyzed explicitly from the theoretical perspective. Moreover, under the Gaussian setting, the Bayes-optimal classifier is actually a linear classifier. Therefore, we believe for linear SVM, the same result as Proposition 1 in our paper will be established.
>
> Reference:
> Sorscher, B., Geirhos, R., Shekhar, S., Ganguli, S., and Morcos, A.S. Beyond neural scaling laws: beating power law scaling via data pruning. NeurIPS 2022.

---

### Official Review · Reviewer_4Hr9 · 2022-10-27

**Confidence:** 4
**Correctness:** 3
**Technical Novelty And Significance:** 2
**Empirical Novelty And Significance:** 2
**Recommendation:** 3

**Clarity, Quality, Novelty And Reproducibility:**

Clarity: The paper is not very clear. There are also some small grammar mistakes like “the bigger … the small(er)”.

Quality: The paper needs more quality figures (legend for Fig. 1 is missing) to explain the method.

Novelty: The idea is sligthly novel.

Reproducibility: The code is uploaded but it is not reproducible. For instance, some hyperparamaters are missing in the article (c, tau etc.).


**Strength And Weaknesses:**

(+)
- The idea of generating more discriminative images to obtain more robustness is powerful and valid.
- The reasoning for selecting new synthetic data in a contrastive framework is valid.

(-)
- The inspiration comes from previous work, the idea is incremental.
- In Table 1, no better results for robust accuracy on “Real data/d=2” is obtained. Also same values are repeated for “Real” and “c=0.5”.
- There is a problem in Algorithm 1. For Lines 5-6, a more clear explanation is needed. At Line 5, z is created and it is not clearly shown how it is used.
- The results are presented for only WRN-28-10. The results should be presented for recent neural network architectures that achieve state-of-the-art performance values on the used datasets.
- Song et al.’s 12th formula is used as the 4th formula in the paper. However no random noise is applied to make it deterministic here. Ablation study for nondeterministic setups should also be included.
-DDIM also used CelebA, LSUNBedroom and LSUNChurch datasets, i.e. not only CIFAR-10 and Traffic Signs. In this work, the authors  should also demonstrate the performance on those relatively more complex datasets.
- Results given in CIFAR-10 are incremental. Also in Table 3, there is nearly no difference between DDPM and Contrastive-DP. Also no std. values are given in the tables. Multiple initializations and training runs should be run and the standard deviations should be reported in the tables.
-Time complexity should be analyzed. Using Contrastive-DP instead of DDIM, there may be a tradeoff between time and accuracy. Is the given incremental results worthy of it?
- Figure 1 is hard to understand. Should be explained clearly in its caption.


**Summary Of The Paper:**

In the paper, Contrastive-Guided Diffusion Process (Contrastive-DP) for robust training is presented. Using Contrastive-DP, a diffusion model to generate new data could be created. For a classification task, using a contrastive setup, better results than DDIM are obtained. The article is heavily inspired by Gowal et al. (2021) and (Song et al., 2021a). The most prominent point is using both real data and synthetic data as in (Carmon et al., 2019) and the effort to select discriminative features. Thus, it can be viewed as an extension to the previous work.

**Summary Of The Review:**

As the method is not explained very clearly, the paper lacks necessary experiments, and explanations, which are explained above, and the results seem to produce incremental gains or none, the contribution of the paper is not adequate for ICLR.

---

> ### Author Response · Authors · 2022-11-18
> **For reviewer 4Hr9**
>
> We thank the reviewer for the comments and suggestions. We appreciate the time you spent on the paper. Below we address the concerns and comments that you have provided.
>
> **Q**:*The inspiration comes from previous work, the idea is incremental.*
>
> **A**: Our work is indeed inspired by Gowal et al. (2021) and Song et al. (2021). The inspiration comes from using synthetic data to improve adversarial robustness. However, our work studies this problem from very different perspectives. We emphasize the key differences and novelties below. First, Gowal et al. (2021) aim to improve the adversarial robustness by utilizing additional training data generated by different generative models. They found that the best performance is achieved by Denoising Diffusion Probabilistic Model (DDPM). On the contrary, we aim to study what kind of synthetic distributions can lead to better performance improvement (and thus improve the sample complexity of adversarial learning), and we presented: (i) theoretical insights for the desired properties of the efficient synthetic data; and (ii) a practical algorithm to enable efficient generation of synthetic data that meets the desired properties. On the other hand, Song et al. (2021) proposed DDIM, which can accelerate the general diffusion generating process. While our work focuses on the sample efficiency of generated data for the downstream task (adversarial learning). Moreover, it is worthwhile mentioning that the proposed generation algorithm in our work is not restricted to DDIM, but any reasonable acceleration techniques can be used. To summarize, we analyze the optimality condition of synthetic distribution for achieving non-trivial robust accuracy and design the guidance for the downstream task. To the best of our knowledge, none of these two directions has been explicitly studied for the adversarial learning task.
>
> **Q**:*In Table 1, no better results for robust accuracy on “Real data/d=2” is obtained. Also same values are repeated for “Real” and “c=0.5”.*
>
> **A**:Under the setting of $\tilde{\boldsymbol{\mu}}=c\boldsymbol{\mu}$, the larger the $c$ is, the better performance we can get on standard accuracy. Our experimental results in Table 1 (together with Table 7-9 in the Appendix) verify the second case in Proposition 1. We also conduct the experiments on $\tilde{\boldsymbol{\mu}}=c (\boldsymbol{\mu} - \varepsilon \mathbf{1}_d)$. Table 10-13 in the Appendix can verify the third case in Proposition 1, i.e., the larger the $c$ is, the better performance we can get on robust accuracy. Moreover, thank you for pointing out the typo in Table 1. The correct results can be found in Table 7 to Table 9.
>
> **Q**:*There is a problem in Algorithm 1. For Lines 5-6, a more clear explanation is needed. At Line 5, z is created and it is not clearly shown how it is used.*
>
> **A**:Sorry for the confusion caused. This was a typo, and we have now deleted line 5 in the revised version.
>
> **Q**:*Song et al.’s 12th formula is used as the 4th formula in the paper. However, no random noise is applied to make it deterministic here. In this work, the authors should also demonstrate the performance on those relatively more complex datasets.*
>
> **A**:Indeed, no random noise is applied in the vanilla DDIM generation framework (corresponding to the case where noise variance equals to $0$). As stated in Song et al. (the paragraph below the 12th formula), the author demonstrates that when $\sigma=0$, it is the exact form of DDIM, which is the same form as the 4th formula in our paper.

---

> ### Author Response · Authors · 2022-11-18
> **For reviewer 4Hr9 (Continued)**
>
> **Q**:*The results are presented for only WRN-28-10. The results should be presented for recent neural network architectures that achieve state-of-the-art performance values on the used datasets. Results given in CIFAR-10 are incremental. Also in Table 3, there is nearly no difference between DDPM and Contrastive-DP. Also no std. values are given in the tables. Multiple initializations and training runs should be run and the standard deviations should be reported in the tables. -Time complexity should be analyzed. Using Contrastive-DP instead of DDIM, there may be a tradeoff between time and accuracy. Is the given incremental results worthy of it?*
>
> **A**:Thank you very much for this suggestion. The numerical experiments were conducted on 4 NVIDIA A100 Tensor Core GPUs, and the running time for DDIM and contrastive-DP is comparable (about two days). Based on our empirical observations, the results were consistent across multiple runs, and we were not able to include the standard error resulting from multiple runs due to limited computing resources. We have reported the results on Traffic Signs dataset for three runs on the revised version (Table 4), and We will add these in the future revised version.
>
> **Q**:*Figure 1 is hard to understand. Should be explained clearly in its caption. The paper needs more quality figures (legend for Fig. 1 is missing) to explain the method.*
>
> **A**:The meaning of Figure 1 is explained clearly in Proposition 1. The notations in Figure 1 are exactly the same as the notation used in Proposition 1. In detail, the yellow circles represent the two Gaussian distributions with mean $\pm \mu$ (underlying true data-generating distribution); and the dashed red circles represent the "non-informative'' synthetic distribution (corresponding to Case 1 in Proposition 1); and the green circles represent the "effective'' synthetic distribution (corresponding to Case 2 and 3) in Proposition 1. Overall,  Figure 1 is not a proposed method, nor a numerical result, but just an illumination of Proposition 1.
>
> **Q**:*The code is uploaded but it is not reproducible. For instance, some hyperparamaters are missing in the article (c, tau etc.).*
>
> **A**:$c$ is not a hyperparameter in real-world experiments. It is only related to the experiments in Table 1 and Table 7-9. All the chosen values are reported in each Table. $\tau$ is a hyperparameter whose value is mentioned in the second paragraph of Appendix B.1.
>
> **Q**:*There are also some small grammar mistakes like “the bigger … the small(er)”.*
>
> **A**:Thank you for pointing out the typo. We have fixed it in the revised version, and we have also run a thorough check throughout the paper for typos, etc.

---

### Decision · Program_Chairs · 2023-01-20

**Decision:**

Reject

**Justification For Why Not Higher Score:**

The experimental evaluation, specifically in the  proposed, simplified setting, is too weak.

**Justification For Why Not Lower Score:**

N/A

**Metareview: Summary, Strengths And Weaknesses:**

This manuscript proposes to use supplementary synthetic, generated data as additional training data to improve generalization, especifically with respect to adversarial examples. Empirical assessment shows improvements when generating synthetic training images in a  guided way,  combined with adversarial training.
While the addressed proble is relevant, the reviews agree that the experiments are too weak in several aspects. For example, the improved results could be due to the increase in training data size and not in particular due to the specific data generation method. Also the discuassion in the context of related approaches is too weak.